# SEER: Transformer-based Robust Time Series Forecasting via Automated Patch Enhancement and Replacement

Xiangfei Qiu [1]  Xvyuan Liu [1]  Tianen Shen [1]  Xingjian Wu [1]  Hanyin Cheng [1]  Bin Yang [1]  Jilin Hu [1 2 3]

## Abstract

Time series forecasting is important in many fields that require accurate predictions for decision-making. Patching techniques, commonly used and effective in time series modeling, help capture temporal dependencies by dividing the data into patches. However, existing patch-based methods fail to dynamically select patches and typically use all patches during the prediction process. In real-world time series, there are often low-quality issues during data collection, such as missing values, distribution shifts, anomalies and white noise, which may cause some patches to contain low-quality information, negatively impacting the prediction results. To address this issue, this study proposes a robust time series forecasting framework called **SEER**. Firstly, we propose an *Augmented Embedding Module*, which improves patch-wise representations using a Mixture-of-Experts (MoE) architecture and obtains series-wise token representations through a channel-adaptive perception mechanism. Secondly, we introduce a *Learnable Patch Replacement Module*, which enhances forecasting robustness and model accuracy through a two-stage process: 1) a dynamic filtering mechanism eliminates negative patch-wise tokens; 2) a replaced attention module substitutes the identified low-quality patches with global series-wise token, further refining their representations through a causal attention mechanism. Comprehensive experimental results demonstrate the SOTA performance of SEER.

[1]School of Data Science and Engineering, ECNU [2]Shanghai Engineering Research Center of Big Data Management [3]National Institutes of Educational Policy Research, ECNU. Correspondence to: Jilin Hu <jlhu@dase.ecnu.edu.cn>.

*Proceedings of the 43rd International Conference on Machine Learning*, Seoul, South Korea. PMLR 306, 2026. Copyright 2026 by the author(s).

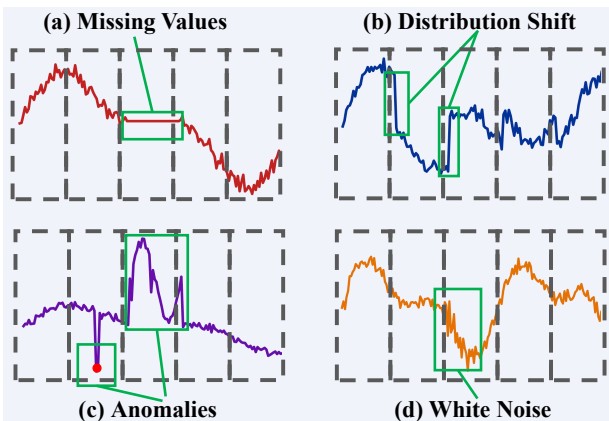

*Figure 1.* Common Scenarios of Low-Quality Datasets.

## 1. Introduction

Time series are generated in diverse domains such as economic (Wu et al., 2024; Wang et al., 2025; Huang et al., 2022), traffic (Qiu et al., 2026; Wang et al., 2026b; Wu et al., 2026), health (Wu et al., 2025a; Wang et al., 2026c; Cheng et al., 2026), energy (Wang et al., 2026a; Liu et al., 2026a), and AIOps (Wang et al., 2023; Qi et al., 2023; Shao et al., 2025). Accurate forecasting of future time series data is crucial for optimizing resource allocation and enhancing decision-making quality (Qiu et al., 2025b;a; Wu et al., 2025c;b;d). In recent years, innovative approaches based on patching have achieved significant improvements in forecasting performance by dividing time series into semantically meaningful patches (Cirstea et al., 2022; Nie et al., 2023), effectively overcoming the limitations of traditional methods in capturing long-range dependencies.

However, in real-world application scenarios—see Figure 1, the time series data collection process often faces multiple challenges: a) missing values may occur during the data collection process due to factors such as sensor malfunctions, communication errors, or interruptions in data transmission. b) the data generation mechanism may evolve over time, causing distributional shift phenomena. c) system failures or unexpected events may introduce anomalies in the data. d) data noise is inevitably introduced during sensor collection or manual recording processes. These complex factors

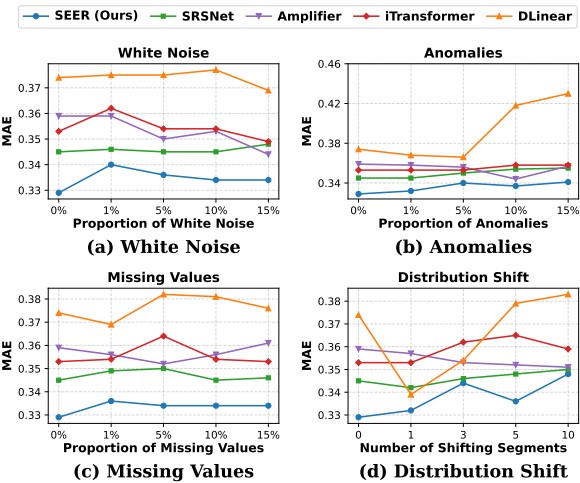

*Figure 2.* To assess SEER's robustness under low-quality data conditions—see Section 3.1, we inject varying proportions of missing values, distribution shifts, anomalies, and white noise into the ETTh2. We evaluate SRSNet, Amplifier, iTransformer, and DLinear as baselines. SRSNet inherently handles non-stationarity, while the others are enhanced with RobustTSF (Cheng et al., 2024) for improved robustness.

result in not all patches containing valid predictive information, with some patches even negatively impacting model performance, ultimately causing model performance deterioration. Therefore, there exists an urgent need to develop robust time-series forecasting methods under low-quality data conditions [1].

In this study, we propose **SEER**, a transformer-based robu**S**t time series forecasting framework via automat**E**d patch **E**nhancement and **R**eplacement. On the one hand, an *Augmented Embedding Module* is designed to enhance both patch-wise and series-wise token representations. At the patch level, this module employs a Mixture-of-Experts (MoE) architecture with adaptive routing to intelligently aggregate patches sharing similar temporal patterns, creating multiple heterogeneous linear representation spaces that enrich patch-wise semantic features. Simultaneously, a channel-adaptive perception mechanism enables series-wise tokens to effectively capture global temporal patterns. On the other hand, the *Learnable Patch Replacement Module* is introduced to enhance the forecasting robustness through a two-stage process. First, a scoring mechanism dynamically identifies and eliminates noisy or uninformative patches. Subsequently, the Replaced Attention module substitutes the identified low-quality patches with global series-wise

---

[1]Low-quality data in this paper refers to non-structural artifacts caused by external factors (e.g., sensor failures or transmission errors), rather than intrinsic temporal dynamics. We acknowledge that, in certain domains such as seismic analysis, anomalies or distribution shifts may themselves carry meaningful signals.

token, further refining their representations through a causal attention mechanism. Finally, we employ a multi-head self-attention mechanism to further refine the patch representations, while utilizing an adaptive dimensionality reduction projection to distill and compress essential information during the prediction stage.

As illustrated in Figure 2, SEER demonstrates significantly smaller performance degradation than baseline methods under various low-quality data scenarios, proving its superior robustness. Please refer to Section 5.3.2 for more extensive experiments and analyses regarding model robustness.

Our contributions are summarized as follows.

- We present SEER, a novel framework designed to enhance prediction robustness through automated patch enhancement and replacement.

- We design an Augmented Embedding Module that improves patch-wise representations using a Mixture-of-Experts architecture and obtains series-wise representations through a channel-adaptive perception mechanism.

- We propose a Learnable Patch Replacement Module that dynamically removes negative patch-wise tokens and fills semantic gaps in filtered patches using globally optimized series-wise tokens.

- We conduct experiments to validate the robustness of SEER on low-quality datasets and its accuracy on multiple commonly used datasets. Additionally, all datasets and code are available at: `https://github.com/decisionintelligence/SEER`.

## 2. Related Works

### 2.1. Patch-based Time Series Forecasting

Patching is a commonly used and effective technique in time series modeling, first introduced by Triformer (Cirstea et al., 2022) and PatchTST (Nie et al., 2023). In Triformer and PatchTST, time series are divided into subsequence-level patches, which are then passed as input tokens to the Transformer, enabling the modeling of temporal dependencies at the patch level. Crossformer (Zhang & Yan, 2022) further extends this idea by segmenting each time series into patches and employing self-attention mechanisms to model dependencies across both the channel and temporal dimensions. xPatch (Stitsyuk & Choi, 2025) adopts the same patch segmentation strategy as PatchTST but introduces a dual-stream architecture consisting of an MLP-based linear stream and a CNN-based nonlinear stream. PatchMLP (Tang & Zhang, 2025) delves deeper into the effectiveness of patches in time series forecasting and employs a multi-scale patch embedding method. However, these methods fail to automatically

select useful patches, treating all patches equally and including them in the final prediction, and they also fail to effectively enrich the representation of patches.

## 2.2. Robust Time Series Forecasting

A series of studies focus on developing robust time series forecasting models under different low-quality data settings: (Yoon et al., 2022) propose enhancing the robustness of forecasting models to input perturbations, motivated by adversarial examples in classification (Ditzler et al., 2015); DUET (Qiu et al., 2025c), Non-stationary Transformers (Liu et al., 2022), RevIN (Kim et al., 2021), and DAIN (Passalis et al., 2020) propose making forecasting models robust to non-stationary time series under the assumptions of concept drift or distribution shift; SRSNet (Wu et al., 2025b) proposes to adaptively select and reassemble patches but lacks an explicit mechanism to tackle the non-robust situations. Merlin (Yu et al., 2025b) and GinAR+ (Yu et al., 2025a) propose enhancing the robustness of forecasting models in datasets with missing values; (Connor et al., 1994; Bohlke-Schneider et al., 2020; Cheng et al., 2024) focus on the robustness of forecasting models in the presence of anomalies in the data. However, most of these methods focus on only one aspect of robustness, resulting in a limited scope. In contrast, our approach addresses forecasting problems in multiple low-quality datasets.

## 3. Preliminaries

### 3.1. Definitions

**Definition 3.1** (Time series). A time series $X \in \mathbb{R}^{N \times T}$ contains $T$ equal-spaced time points with $N$ channels. If $N = 1$, a time series is called univariate, and multivariate if $N > 1$. For clarity, we denote $X_{i,j}$ as the $j$-th time point of $i$-th channel.

**Definition 3.2** (White Noise). Given a time series $\mathcal{X} \in \mathbb{R}^{N \times L}$, it may be affected by the error from the signal source, often imposing white noise $\epsilon \sim \mathcal{N}(\mu, \sigma^2)$ to the actual values, which may hinder the forecasting performance.

**Definition 3.3** (Anomalies). For time series $\mathcal{X} \in \mathbb{R}^{N \times L}$ from cyber-physical systems, there exists anomalies due to the instabilities of monitoring targets or the systems themselves. However, there exists no labels to indicate whether $\mathcal{X}_{i,j}$ is an anomaly or not in forecasting tasks.

**Definition 3.4** (Missing Values). Since a time series $\mathcal{X} \in \mathbb{R}^{N \times L}$ considers the equal-spaced case, this causes difficulties in collecting the values of each time point in real-world applications. And if $\mathcal{X}_{i,j}$ misses, the value is set to 0.

**Definition 3.5** (Distribution Shift (Qiu et al., 2025c)). Given a time series $\mathcal{X} \in \mathbb{R}^{N \times L}$, a set of time series segments with length $T$ is obtained through window sliding, denoted as $\mathcal{D} = \{\mathcal{X}_{n,i:i+T} | n \in [1, N] \text{ and } i \in [1, L-T]\}$, where each

$\mathcal{X}_{n,i:i+T}$ is equivalent to $X_{n,:}$. Distribution shift occurs when $\mathcal{D}$ can be clustered into $K$ subsets, i.e., $\mathcal{D} = \bigcup_{i=1}^{K} \mathcal{D}_i$. Each subset $\mathcal{D}_i$ corresponds to a distinct data distribution $P_{\mathcal{D}_i}(x)$, satisfying $P_{\mathcal{D}_i}(x) \neq P_{\mathcal{D}_j}(x)$ for any $i \neq j$ within $1 \leq i, j \leq k$.

### 3.2. Problem Statement

**Multivariate Time Series Forecasting** aims to utilize the historical time series $X = \langle X_{:,1}, \cdots, X_{:,T} \rangle \in \mathbb{R}^{N \times T}$ with $N$ channels and $T$ timestamps to predict the next $F$ future timestamps, formulated as $Y = \langle X_{:,T+1}, \cdots, X_{:,T+F} \rangle \in \mathbb{R}^{N \times F}$. **Robust Multivariate Time Series Forecasting** refers to making forecasts on Multivariate Time Series with White Noise (Def 3.2), Anomalies (Def 3.3), Missing Values (Def 3.4) or Distribution Shift (Def 3.5).

## 4. Methodology

### 4.1. Overall Framework

Figure 3 shows the framework of SEER, which enhances both algorithmic accuracy and robustness through dual-scale (patch-wise and series-wise) representation enhancement and a learnable dynamic patch replacement mechanism. Specifically, we first employ *Instance Normalization* (Kim et al., 2021) to standardize the distribution of training and testing data. We then design a *Augmented Embedding Module* consisting of two components: the Augmented Patch Embedding component (Figure 3a) utilizes a Mixture of Experts (MoE) architecture that adaptively aggregates patches with similar patterns through a routing mechanism to form multiple heterogeneous linear representation spaces, thereby enhancing the semantics of patch-wise tokens, while the Augmented Series Embedding component (Figure 3b) integrates a channel-adaptive perception mechanism to capture global temporal patterns in time series. We further design the *Learnable Patch Replacement Module*, which enhances forecasting robustness through a two-stage process where a scoring mechanism (Figure 3c) first dynamically identifies and eliminates noisy or uninformative patches, followed by a replaced causal self-attention mechanism (Figure 3d) to refine their representations. The framework concludes with a standard Multi-Head Self-Attention layer followed by a linear predictor for future value forecasting.

### 4.2. Augmented Embedding Module

#### 4.2.1. AUGMENTED PATCH EMBEDDING

Due to the continuity of time series data, it is difficult to discretize and tokenize them while keeping enough representational capabilities. This problem is especially crucial in non-robust time series, where insufficient representational capabilities may confuse normal and abnormal time series

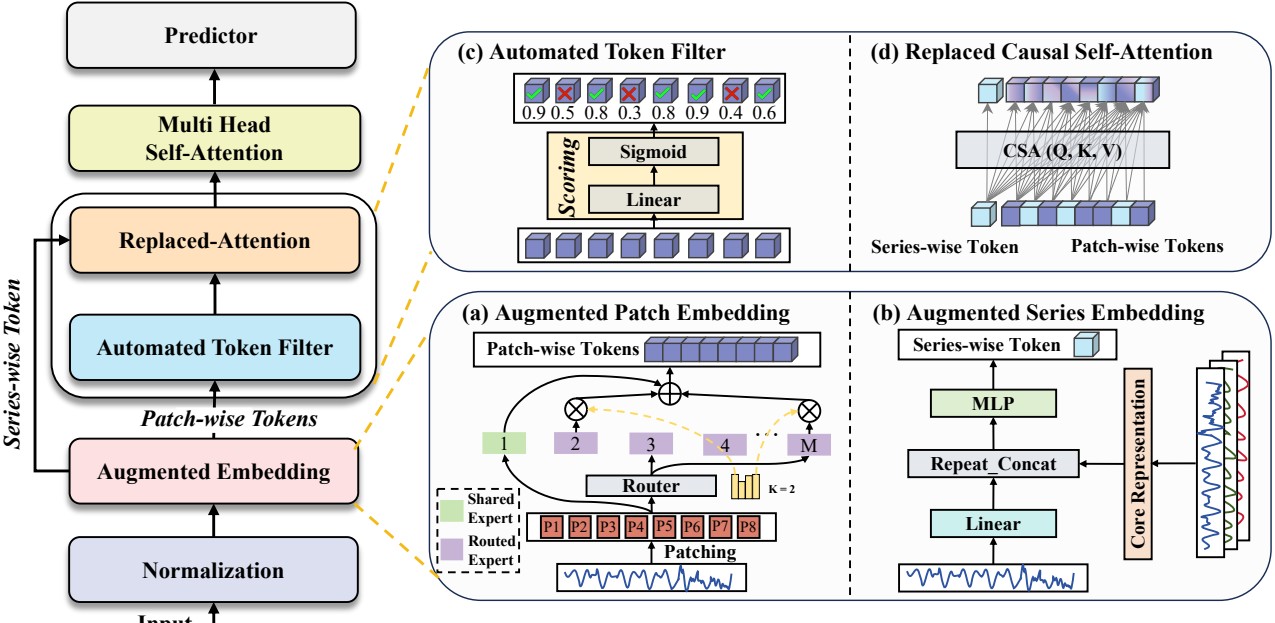

*Figure 3.* The overall architecture of SEER primarily comprises the Augmented Embedding Module and the Learnable Patch Replacement Module, the latter of which integrates Automated Token Filtering and Replaced Attention mechanisms.

tokens and model them in the same hidden space, which hinders the forecasting performance. Unfortunately, most recent studies adopt the Conventional Patch Embedding (Nie et al., 2023), which utilizes a single linear projection to produce value embeddings, thereby mapping time series tokens into a linear space.

Given a time series $X \in \mathbb{R}^{N \times T}$, it is first divided into patches: $X^P \in \mathbb{R}^{N \times n \times p}$, where $p$ is the patch size and $n$ is the number of patches. To augment the embeddings in Robust Time Series Forecasting tasks, we utilize the Mixture-of-Experts to enhance the representational capabilities, which can construct heterogeneous embedding spaces through routing different experts, thus enhancing the semantics of tokens. Specifically, we define each expert as the conventional linear value embedding:

$$\text{expert}(X^P) = \text{Linear}(X^P) \in \mathbb{R}^{(N \times n) \times d}, \quad (1)$$

where $d$ denotes the hidden dimension. We utilize the Noisy Gating (Shazeer et al., 2017) to route experts for time series tokens $X^P \in \mathbb{R}^{(N \times n) \times p}$:

$$G(X^P) = \text{Softmax}(\text{KeepTopK}(H(X^P)), k), \quad (2)$$

$$H(X^P) = \text{Linear}_\mu(X^P) + \epsilon \odot \text{Linear}_\sigma(X^P), \quad (3)$$

$$\text{KeepTopK}(\mathcal{V})_i = \begin{cases} \mathcal{V}_i & \text{if } i \in \text{ArgTopk}(\mathcal{V}) \\ -\infty & \text{otherwise} \end{cases}, \quad (4)$$

where $G(X^P) \in \mathbb{R}^{(N \times n) \times k}$ denotes the routing weights of the Top $k$ experts, $\epsilon \sim \mathcal{N}(0, 1)$ is utilized to promote the

stability of training and enhance diversity, known as "noisy" gating. Then the augmented embeddings can be obtained through:

$$X^{P'} = \sum_{i=1}^{\mathcal{M}} \text{expert}_s^i(X^P) + \sum_{i=1}^{k} G(X^P)^i \odot \text{expert}_r^i(X^P), \quad (5)$$

where we also adopt $\mathcal{M}$ shared experts $\text{expert}_s^i$ to extract the general patterns, and select $k$ experts from total $M$ routed experts $\text{expert}_r^i$ for each token. The augmented time series embeddings are $X^{P'} \in \mathbb{R}^{(N \times n) \times d}$.

### 4.2.2. AUGMENTED SERIES EMBEDDING

Our primary motivation is to enhance the patch embeddings and replace ones having potential low-quality information which causes instability in time series forecasting. Intuitively, the prototype of the substitute needs meticulous designs to well replace the non-stable patch embeddings.

Inspired by SOFTS (Han et al., 2024), which utilizes a star structure to obtain a series-wise token, we adopt their method to construct high-quality prototypes of the substitutes. We obtain the embeddings of all channels through:

$$X^E = \text{Linear}(X), \quad (6)$$

where $X^E \in \mathbb{R}^{N \times d}$ are the channel embeddings. We then utilize the stochastic pooling (Zeiler & Fergus, 2013) technique to get the core representation $o$ by aggregating repre-

sentations of $N$ series:

$$o = \text{Stoch\_Pool}(\text{MLP}_1(X^E)), \qquad (7)$$

where $\text{MLP}_1$ projects $X^E$ to dimension of $\mathbb{R}^{N \times d'}$ and the stochastic pooling makes a reduction operation to obtain the series-wise token (prototype) with the dimension of $\mathbb{R}^{d'}$. Then we extend the global prototype to series-wise prototypes to enhance the representational capability:

$$F = \text{Repeat\_Concat}(X^E, o), O = \text{MLP}_2(F), \quad (8)$$

where $O \in \mathbb{R}^{N \times d}$ are the series-wise prototypes of substitutes and we use them in the subsequent Learnable Patch Replacement Module to replace the non-stable tokens. The Repeat_Concat broadcasts along the channel dimension to concat $o$ to each channel of $X^E$. $\text{MLP}_2$ is used to align the dimension of $F$ with $X^E$.

### 4.3. Learnable Patch Replacement Module

#### 4.3.1. AUTOMATED TOKEN FILTER

After the Augmented Embedding Module generates the enhanced time series tokens and prepares the series-wise prototypes for replacement. We then manage to filter out the time series tokens to enhance the robustness. Specifically, we introduce an Automated Token Filter to evaluate the quality of each token, then choose to preserve or filter them:

$$\text{scores} = \text{Sigmoid}(\text{Linear}(X^{P'})), \mathcal{I} = (\text{scores} > \tau), \quad (9)$$

$$\mathcal{F} = \text{scores}, \mathcal{F}_{inv} = 1/\text{detach}(\mathcal{F}), \qquad (10)$$

$$\text{Identity} = \mathcal{F} \odot \mathcal{F}_{inv}, \qquad (11)$$

where the Automated Token Filter first generates scores $\in \mathbb{R}^{N \times n}$ for tokens through a linear projection and the sigmoid activation function to limit the scores in $[0, 1]$. Then it generates the indicators $\mathcal{I} \in \mathbb{R}^{N \times n}$ of tokens based on their scores compared with predefined threshold $\tau$, where $\mathcal{I}_{i,j} = 0$ indicates the $j$-th token in $i$-th channel is filtered, and $\mathcal{I}_{i,j} = 1$ indicates preservation. To keep the propagation of gradients, we make a hard connection in Formula (11), creating an all 1 Identity $\in \mathbb{R}^{N \times n}$ matrix with gradients attached for subsequent mask generation. Under this mechanism, we can support filtering out different numbers of tokens in distinct channels, which enhances the flexibility.

#### 4.3.2. REPLACED CAUSAL SELF-ATTENTION

This module aims to substitute identified low-quality patches with a global series-wise prototype, subsequently refining their representations via a causal attention mechanism.

First, to maintain gradient back-propagation through the discrete filtering decision, we first compute the gradient-aware mask $\mathcal{M} \in \mathbb{R}^{N \times n}$ by coupling the binary indicators

$\mathcal{I}$ with the identity matrix:

$$\mathcal{M} = \mathcal{I} \odot \text{Identity}. \qquad (12)$$

Based on this mask, we implement a replacement operation. For each channel $i$ and patch index $j$, the augmented patch embedding $X_{i,j}^{P'}$ is substituted by its corresponding series-wise prototype $O_i$ if the filter deems the original patch to be of low quality:

$$X_{i,j}^{F'} = X_{i,j}^{P'} \odot \mathcal{M}_{i,j} + O_i \odot (1 - \mathcal{M}_{i,j}), \qquad (13)$$

where $X^{F'} \in \mathbb{R}^{N \times n \times d}$ represents the replaced sequence.

To further capture temporal dependencies and allow global information to guide local patch representations, we prepend the series-wise prototype $O$ as a global token to the sequence, resulting in the augmented input $\hat{X}^F \in \mathbb{R}^{N \times (n+1) \times d}$. We then apply Multi-head Causal Self-Attention (MCSA) to model the interactions:

$$X^{out} = \text{MCSA}(\hat{X}^F, \hat{X}^F, \hat{X}^F, M_{mask}), \qquad (14)$$

where $M_{mask} \in \mathbb{R}^{(n+1) \times (n+1)}$ is the causal mask. This mask ensures that each patch attends only to its preceding patches and the prepended global prototype. Consequently, even when multiple non-robust patches are replaced by the same series-wise token $O$, their resulting representations in $X^{out}$ remain distinct. This is because causal attention enables each position to aggregate a unique historical context, thereby preserving temporal continuity and position-specific semantics of the original time series.

Following the Learnable Patch Replacement Module, we utilize a standard Multi-head Self-Attention (MSA) layer to facilitate deeper integration between the global prototype and the patch tokens:

$$X^O = \text{MSA}(X^{out}, X^{out}, X^{out}) \in \mathbb{R}^{N \times (n+1) \times d}. \quad (15)$$

### 4.4. Predictor

The predictor is designed to efficiently transform the refined token representations into final forecasting outputs.

We have empirical observations that reducing the dimension of $X^O$ can compress the information in tokens and preserve the performance, which is beneficial for efficiency:

$$X^{O'} = \text{Linear}(X^O), \qquad (16)$$

where the compressed $X^{O'} \in \mathbb{R}^{N \times (n+1) \times \tilde{d}}$ and $\tilde{d} << d$. Finally, we adopt the commonly-used Linear Flatten Head to predict the future values:

$$\hat{Y} = \text{Linear}(\text{Flatten}(X^{O'})), \qquad (17)$$

where $\hat{Y} \in \mathbb{R}^{N \times L}$ is the forecasting results over the horizon of length $L$.

*Table 1.* Statistics of datasets.

| Dataset | Domain | Frequency | Lengths | Dim | Split | Description |
|---|---|---|---|---|---|---|
| ETTh1 | Electricity | 1 hour | 14,400 | 7 | 6:2:2 | Power transformer 1, comprising seven indicators such as oil temperature and useful load |
| ETTh2 | Electricity | 1 hour | 14,400 | 7 | 6:2:2 | Power transformer 2, comprising seven indicators such as oil temperature and useful load |
| ETTm1 | Electricity | 15 mins | 57,600 | 7 | 6:2:2 | Power transformer 1, comprising seven indicators such as oil temperature and useful load |
| ETTm2 | Electricity | 15 mins | 57,600 | 7 | 6:2:2 | Power transformer 2, comprising seven indicators such as oil temperature and useful load |
| Weather | Environment | 10 mins | 52,696 | 21 | 7:1:2 | Recorded every for the whole year 2020, which contains 21 meteorological indicators |
| Electricity | Electricity | 1 hour | 26,304 | 321 | 7:1:2 | Electricity records the electricity consumption in kWh every 1 hour from 2012 to 2014 |
| Solar | Energy | 10 mins | 52,560 | 137 | 6:2:2 | Solar production records collected from 137 PV plants in Alabama |
| Traffic | Traffic | 1 hour | 17,544 | 862 | 7:1:2 | Road occupancy rates measured by 862 sensors on San Francisco Bay area freeways |

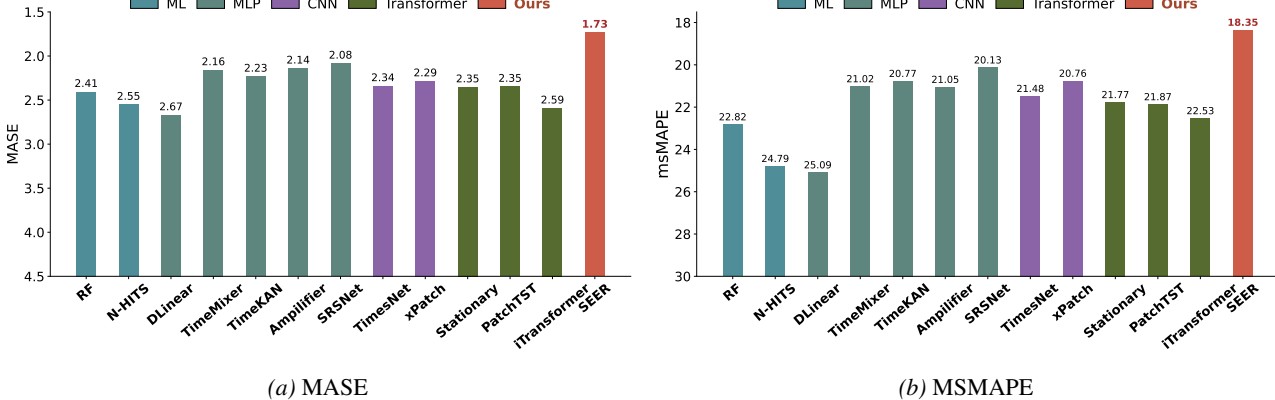

*(a) MASE*  *(b) MSMAPE*

*Figure 4.* Univariate forecasting results.

*Table 2.* Statistics of univariate datasets.

| Frequency | #Series | Seasonality | Trend | Shifting | Transition | Stationarity | $|TS| < 300$[1] | $F$[2] |
|---|---|---|---|---|---|---|---|---|
| Yearly | 1,500 | 611 | 1,086 | 978 | 633 | 354 | 1,499 | 6 |
| Quarterly | 1,514 | 486 | 933 | 889 | 894 | 471 | 1,508 | 8 |
| Monthly | 1,674 | 883 | 884 | 778 | 1,212 | 667 | 1,026 | 18 |
| Weekly | 805 | 253 | 330 | 445 | 407 | 372 | 473 | 13 |
| Daily | 1,484 | 374 | 502 | 487 | 1,176 | 714 | 442 | 14 |
| Hourly | 706 | 435 | 276 | 284 | 680 | 472 | 0 | 48 |
| Other | 385 | 75 | 248 | 236 | 195 | 124 | 215 | 8 |
| Total | 8,068 | 3,117 | 4,259 | 4,097 | 5,197 | 3,174 | 5,163 | |

[1] $|TS| < 300$: the length of univariate time series <300

[2] $F$: the forecasting horizon

## 5. Experiments

### 5.1. Experimental Settings

#### 5.1.1. DATASETS

To conduct comprehensive comparisons among different models, we evaluate the proposed method on multiple forecasting settings. 1) For multivariate time series forecasting, we conduct experiments on eight widely used benchmark datasets, including ETT (four subsets), Weather, Electricity, Solar, and Traffic. 2) For univariate time series forecasting, we utilize 8,068 univariate time series collected from the TFB (Qiu et al., 2024) benchmark, enabling large-scale and comprehensive evaluations across diverse temporal patterns. 3) For robust time series forecasting, we evaluate model robustness under different corruption scenarios. Specifically, ILI and NASDAQ from TFB are used to evaluate robustness under distribution shifts; FRED-MD and Wiki2000 are adopted for white noise scenarios; SMD and PSM from

TAB (Qiu et al., 2025a) are used for anomaly corruption; and USHCN together with Human Activity are employed to evaluate forecasting performance under missing values. Tables 1 and 2 present the statistical information of the multivariate and univariate datasets, respectively.

#### 5.1.2. BASELINES

We conduct a comprehensive comparison of our model against multiple baselines, including the latest state-of-the-art (SOTA) models. These baselines include the 2025 SOTA models: SRSNet (Wu et al., 2025b), DUET (Qiu et al., 2025c), xPatch (Huang et al., 2025), and Amplifier (Fei et al., 2025), as well as the 2024 and 2023 SOTA models: iTransformer (Liu et al., 2024), PatchTST (Nie et al., 2023), Fredformer (Piao et al., 2024), and DLinear (Zeng et al., 2023). We also include some classic baselines in univariate experiments, e.g., RF (Mei et al., 2014), TimeMixer (Wang et al., 2024), Non-stationary Transformer (Liu et al., 2022), and FEDFormer (Zhou et al., 2022).

#### 5.1.3. IMPLEMENTATION DETAILS

To keep consistent with previous works, we adopt Mean Squared Error (mse) and Mean Absolute Error (mae) as evaluation metrics. For multivariate forecasting, we consider four forecasting horizons $F$: {96, 192, 336, 720} for all the datasets. The look-back window length is fixed at 96. We utilize the TFB (Qiu et al., 2024) code repository for uni-

*Table 3.* Multivariate forecasting full results with forecasting horizons $F \in \{96, 192, 336, 720\}$, the look-back window length is set to 96 for all baselines. Avg means the average results from all four forecasting horizons. **Red**: the best, Blue: the 2nd best.

| Models | | SEER (ours) | | SRSNet (2025) | | DUET (2025) | | xPatch (2025) | | Amplifier (2025) | | FredFormer (2024) | | iTransformer (2024) | | PatchTST (2023) | | DLinear (2023) | |
|---|---|---|---|---|---|---|---|---|---|---|---|---|---|---|---|---|---|---|---|
| Metrics | | mse | mae | mse | mae | mse | mae | mse | mae | mse | mae | mse | mae | mse | mae | mse | mae | mse | mae |
| ETTh1 | 96 | **0.366** | **0.385** | 0.383 | 0.395 | 0.377 | 0.393 | 0.378 | 0.390 | 0.376 | 0.393 | 0.378 | 0.395 | 0.386 | 0.405 | 0.414 | 0.419 | 0.397 | 0.412 |
| | 192 | **0.417** | **0.416** | 0.433 | 0.422 | 0.429 | 0.425 | 0.433 | 0.420 | 0.442 | 0.430 | 0.435 | 0.424 | 0.441 | 0.436 | 0.460 | 0.445 | 0.446 | 0.441 |
| | 336 | **0.454** | **0.435** | 0.476 | 0.446 | 0.471 | 0.446 | 0.484 | 0.445 | 0.478 | 0.446 | 0.485 | 0.447 | 0.487 | 0.458 | 0.501 | 0.466 | 0.489 | 0.467 |
| | 720 | **0.456** | **0.459** | 0.474 | 0.471 | 0.496 | 0.480 | 0.480 | 0.462 | 0.501 | 0.479 | 0.496 | 0.472 | 0.503 | 0.491 | 0.500 | 0.488 | 0.513 | 0.510 |
| | avg | **0.423** | **0.424** | 0.442 | 0.433 | 0.443 | 0.436 | 0.444 | 0.429 | 0.449 | 0.437 | 0.448 | 0.435 | 0.454 | 0.448 | 0.469 | 0.455 | 0.461 | 0.458 |
| ETTh2 | 96 | **0.281** | **0.329** | 0.296 | 0.345 | 0.296 | 0.345 | 0.287 | 0.332 | 0.298 | 0.347 | 0.291 | 0.342 | 0.297 | 0.349 | 0.302 | 0.348 | 0.340 | 0.394 |
| | 192 | **0.358** | **0.379** | 0.370 | 0.394 | 0.368 | 0.389 | 0.360 | 0.382 | 0.378 | 0.401 | 0.372 | 0.390 | 0.380 | 0.400 | 0.388 | 0.400 | 0.482 | 0.479 |
| | 336 | **0.401** | **0.413** | 0.413 | 0.425 | 0.411 | 0.422 | 0.417 | 0.421 | 0.428 | 0.437 | 0.419 | 0.431 | 0.428 | 0.432 | 0.426 | 0.433 | 0.591 | 0.541 |
| | 720 | **0.384** | **0.417** | 0.425 | 0.444 | 0.412 | 0.434 | 0.412 | 0.433 | 0.452 | 0.460 | 0.431 | 0.450 | 0.427 | 0.445 | 0.431 | 0.446 | 0.839 | 0.661 |
| | avg | **0.356** | **0.384** | 0.376 | 0.402 | 0.372 | 0.398 | 0.369 | 0.392 | 0.389 | 0.411 | 0.378 | 0.403 | 0.383 | 0.407 | 0.387 | 0.407 | 0.563 | 0.519 |
| ETTm1 | 96 | **0.308** | **0.337** | 0.319 | 0.358 | 0.324 | 0.354 | 0.316 | 0.343 | 0.318 | 0.356 | 0.326 | 0.361 | 0.334 | 0.368 | 0.329 | 0.367 | 0.346 | 0.374 |
| | 192 | 0.361 | **0.369** | **0.359** | 0.381 | 0.369 | 0.379 | 0.369 | 0.369 | 0.362 | 0.381 | 0.363 | 0.384 | 0.377 | 0.391 | 0.367 | 0.385 | 0.382 | 0.391 |
| | 336 | **0.391** | **0.391** | 0.391 | 0.404 | 0.404 | 0.402 | 0.401 | 0.392 | 0.393 | 0.404 | 0.395 | 0.406 | 0.426 | 0.420 | 0.399 | 0.410 | 0.410 | 0.415 |
| | 720 | **0.450** | **0.429** | 0.470 | 0.436 | 0.463 | 0.437 | 0.461 | 0.429 | 0.460 | 0.442 | 0.456 | 0.441 | 0.491 | 0.459 | 0.454 | 0.439 | 0.473 | 0.451 |
| | avg | **0.377** | **0.381** | 0.385 | 0.395 | 0.390 | 0.393 | 0.387 | 0.383 | 0.383 | 0.396 | 0.385 | 0.398 | 0.407 | 0.410 | 0.387 | 0.400 | 0.404 | 0.408 |
| ETTm2 | 96 | **0.172** | **0.249** | 0.181 | 0.267 | 0.174 | 0.255 | 0.174 | 0.252 | 0.178 | 0.261 | 0.177 | 0.258 | 0.180 | 0.264 | 0.175 | 0.259 | 0.193 | 0.293 |
| | 192 | **0.238** | **0.294** | 0.243 | 0.306 | 0.243 | 0.302 | 0.240 | 0.297 | 0.243 | 0.303 | 0.243 | 0.301 | 0.250 | 0.309 | 0.241 | 0.302 | 0.284 | 0.361 |
| | 336 | **0.297** | **0.334** | 0.306 | 0.346 | 0.304 | 0.341 | 0.302 | 0.335 | 0.305 | 0.344 | 0.311 | 0.348 | 0.302 | 0.340 | 0.305 | 0.343 | 0.382 | 0.429 |
| | 720 | 0.395 | **0.390** | 0.407 | 0.399 | 0.399 | 0.397 | 0.403 | 0.393 | **0.393** | 0.397 | 0.404 | 0.398 | 0.412 | 0.407 | 0.402 | 0.400 | 0.558 | 0.525 |
| | avg | **0.276** | **0.317** | 0.284 | 0.329 | 0.280 | 0.324 | 0.280 | 0.319 | 0.280 | 0.326 | 0.281 | 0.324 | 0.288 | 0.332 | 0.281 | 0.326 | 0.354 | 0.402 |
| Weather | 96 | **0.162** | **0.198** | 0.167 | 0.214 | 0.163 | 0.202 | 0.166 | 0.202 | 0.165 | 0.210 | 0.163 | 0.207 | 0.174 | 0.214 | 0.177 | 0.218 | 0.195 | 0.252 |
| | 192 | 0.211 | **0.240** | 0.215 | 0.255 | 0.218 | 0.252 | **0.210** | 0.242 | 0.212 | 0.253 | 0.224 | 0.258 | 0.221 | 0.254 | 0.225 | 0.259 | 0.237 | 0.295 |
| | 336 | 0.268 | **0.284** | 0.270 | 0.294 | 0.274 | 0.294 | **0.267** | **0.284** | **0.267** | 0.293 | 0.278 | 0.298 | 0.278 | 0.296 | 0.278 | 0.297 | 0.282 | 0.331 |
| | 720 | **0.344** | 0.336 | 0.346 | 0.344 | 0.349 | 0.343 | **0.344** | **0.335** | **0.344** | 0.342 | 0.357 | 0.350 | 0.358 | 0.349 | 0.354 | 0.348 | 0.345 | 0.382 |
| | avg | **0.246** | **0.265** | 0.250 | 0.277 | 0.251 | 0.273 | 0.247 | 0.266 | 0.247 | 0.275 | 0.256 | 0.278 | 0.258 | 0.278 | 0.259 | 0.281 | 0.265 | 0.315 |
| Solar | 96 | **0.185** | **0.199** | 0.216 | 0.258 | 0.200 | 0.207 | 0.234 | 0.245 | 0.186 | 0.232 | 0.189 | 0.236 | 0.203 | 0.237 | 0.234 | 0.286 | 0.290 | 0.378 |
| | 192 | **0.220** | **0.222** | 0.247 | 0.280 | 0.228 | 0.233 | 0.265 | 0.262 | 0.231 | 0.264 | 0.227 | 0.259 | 0.233 | 0.261 | 0.267 | 0.310 | 0.320 | 0.398 |
| | 336 | 0.238 | **0.233** | 0.268 | 0.294 | 0.262 | 0.244 | 0.301 | 0.280 | **0.234** | 0.263 | 0.243 | 0.286 | 0.248 | 0.273 | 0.290 | 0.315 | 0.353 | 0.415 |
| | 720 | 0.241 | **0.241** | 0.268 | 0.290 | 0.258 | 0.249 | 0.308 | 0.284 | **0.238** | 0.265 | 0.250 | 0.285 | 0.249 | 0.275 | 0.289 | 0.317 | 0.357 | 0.413 |
| | avg | **0.221** | **0.224** | 0.250 | 0.281 | 0.237 | 0.233 | 0.277 | 0.268 | 0.222 | 0.256 | 0.227 | 0.267 | 0.233 | 0.262 | 0.270 | 0.307 | 0.330 | 0.401 |
| Electricity | 96 | **0.142** | **0.225** | 0.161 | 0.252 | 0.145 | 0.233 | 0.160 | 0.244 | 0.149 | 0.245 | 0.148 | 0.242 | 0.148 | 0.240 | 0.195 | 0.285 | 0.210 | 0.302 |
| | 192 | **0.158** | **0.240** | 0.172 | 0.261 | 0.163 | 0.248 | 0.169 | 0.253 | 0.165 | 0.260 | 0.165 | 0.257 | 0.162 | 0.253 | 0.199 | 0.289 | 0.210 | 0.305 |
| | 336 | **0.174** | **0.257** | 0.190 | 0.279 | 0.175 | 0.262 | 0.185 | 0.268 | 0.176 | 0.271 | 0.180 | 0.274 | 0.178 | 0.269 | 0.215 | 0.305 | 0.223 | 0.319 |
| | 720 | **0.201** | **0.282** | 0.231 | 0.313 | 0.204 | 0.291 | 0.221 | 0.300 | 0.204 | 0.296 | 0.218 | 0.305 | 0.225 | 0.317 | 0.256 | 0.337 | 0.258 | 0.350 |
| | avg | **0.169** | **0.251** | 0.189 | 0.276 | 0.172 | 0.259 | 0.184 | 0.266 | 0.174 | 0.268 | 0.178 | 0.270 | 0.178 | 0.270 | 0.216 | 0.304 | 0.225 | 0.319 |
| Traffic | 96 | **0.392** | **0.233** | 0.471 | 0.295 | 0.407 | 0.252 | 0.475 | 0.279 | 0.450 | 0.295 | 0.403 | 0.274 | 0.395 | 0.268 | 0.544 | 0.359 | 0.650 | 0.396 |
| | 192 | **0.417** | **0.240** | 0.480 | 0.300 | 0.431 | 0.262 | 0.486 | 0.277 | 0.489 | 0.311 | 0.429 | 0.289 | 0.417 | 0.276 | 0.540 | 0.354 | 0.598 | 0.370 |
| | 336 | **0.430** | **0.247** | 0.496 | 0.306 | 0.456 | 0.269 | 0.500 | 0.280 | 0.484 | 0.321 | 0.441 | 0.295 | 0.433 | 0.283 | 0.551 | 0.358 | 0.605 | 0.373 |
| | 720 | **0.448** | **0.261** | 0.531 | 0.328 | 0.509 | 0.292 | 0.537 | 0.295 | 0.517 | 0.333 | 0.463 | 0.300 | 0.467 | 0.302 | 0.586 | 0.375 | 0.645 | 0.394 |
| | avg | **0.422** | **0.245** | 0.494 | 0.307 | 0.451 | 0.269 | 0.500 | 0.283 | 0.485 | 0.315 | 0.434 | 0.289 | 0.428 | 0.282 | 0.555 | 0.362 | 0.625 | 0.383 |
| $1^{st}$ Count | | **34** | **39** | 1 | 0 | 0 | 0 | 3 | 2 | 5 | 0 | 0 | 0 | 0 | 0 | 0 | 0 | 0 | 0 |

fied evaluation, with all baseline results also derived from TFB. Regarding univariate forecasting, we strictly follow the input and output lengths specified in the TFB framework for all 8,068 univariate time series. Finally, we report the average results across all univariate time series. Following the settings in TFB (Qiu et al., 2024) and TSFM-Bench (Li et al., 2025b), we do not apply the "Drop Last" trick to ensure a fair comparison. All experiments of SEER are conducted using PyTorch (Paszke et al., 2019) in Python 3.8 and executed on an NVIDIA Tesla-A800 GPU. The training process is guided by the L1 loss function and employs the ADAM optimizer. The initial batch size is set to 64, with the flexibility to halve it (down to a minimum of 8) in case of an Out-Of-Memory (OOM) issue.

### 5.2. Main Results

**Univariate Forecasting:** Figure 4 presents the MASE results for univariate time series forecasting on the TFB benchmark, encompassing 8,068 short-term univariate series. We observe that SEER consistently outperforms state-of-the-art baselines, including xPatch, SRSNet, and TimeKAN. In these cases, short time series often possess high uncertainty and volatility, where robustness is very important in training. SEER can adaptively replace the non-robust patches with learnable robust ones, thus mitigating the overfitting phenomena when training on such short series.

**Multivariate Forecasting:** Comprehensive forecasting results are presented in Table 3 with various forecasting hori-

*Table 4.* Ablation study for SEER.

| Datasets | ETTh2 | | ETTm2 | | Weather | | Solar | |
|---|---|---|---|---|---|---|---|---|
| Metrics | mse | mae | mse | mae | mse | mae | mse | mae |
| w/o MOE | 0.365 | 0.389 | 0.280 | 0.320 | 0.249 | 0.269 | 0.225 | 0.226 |
| Simple Series Embedding | 0.379 | 0.396 | 0.279 | 0.321 | 0.254 | 0.271 | 0.240 | 0.232 |
| w/o Token Filter | 0.367 | 0.390 | 0.279 | 0.319 | 0.250 | 0.270 | 0.225 | 0.229 |
| w/o Feature Reduction | 0.372 | 0.394 | 0.282 | 0.322 | 0.249 | 0.268 | 0.228 | 0.227 |
| SEER | **0.356** | **0.384** | **0.276** | **0.317** | **0.246** | **0.265** | **0.221** | **0.224** |

*Table 5.* Robustness comparison of different models under various synthetically perturbed scenarios. All reported results represent the average performance under varying levels of perturbation. Full results are available in Table 7 of Appendix C.1.

| Models | White Noise | | Anomalies | | Missing Value | | Distribution Shift | |
|---|---|---|---|---|---|---|---|---|
| Metrics | mse | mae | mse | mae | mse | mae | mse | mae |
| DLinear | 0.318 | 0.374 | 0.344 | 0.396 | 0.326 | 0.377 | 0.310 | 0.364 |
| iTransformer | 0.310 | 0.355 | 0.311 | 0.356 | 0.310 | 0.356 | 0.316 | 0.360 |
| Amplifier | 0.302 | 0.352 | 0.306 | 0.354 | 0.307 | 0.356 | 0.306 | 0.353 |
| SRSNet | 0.297 | 0.346 | 0.302 | 0.351 | 0.299 | 0.348 | 0.303 | 0.347 |
| SEER | **0.293** | **0.336** | **0.302** | **0.338** | **0.290** | **0.335** | **0.296** | **0.340** |

zons $F \in \{96, 192, 336, 720\}$, with the look-back window set to 96 for all baselines. The best results are highlighted in **Red** and the second-best in Blue. Based on these results, we have the following observations:

1) According to the description in TFB (Qiu et al., 2024), these common datasets are not stationary, and many of them exhibit shifting phenomena, which means that some subsequences may have low-quality segments. From the results, it can be seen that SEER performs well in most cases.

2) SEER demonstrates strong predictive performance across multiple forecasting horizons. From an absolute performance perspective, SEER shows improvement over the second-best baseline model, xPatch, with a 7.3% reduction in MSE and a 4.9% reduction in MAE. Additionally, SEER achieved first place in 34 out of 45 MSE settings and in 39 out of 45 MAE settings.

## 5.3. Model Analyses

### 5.3.1. ABLATION STUDIES

We compare the full version of SEER with several variants to examine the contributions of each component. Table 4 illustrates the unique impact of each module. We have the following observations: 1) *After removing MOE*, compared to SEER, the performance of the algorithm slightly decreases on both lower-quality datasets (such as ETTh2 and ETTm2) and higher-quality datasets (such as Weather and Solar). This indicates that the Mixture-of-Experts technique enhances patch-wise representations, especially in datasets with more complex temporal patterns (like ETTh2 and ETTm2), improving overall model performance. 2) *Only use simple series embedding*: When the augmented

series embeddings are replaced by a simple series embedding—where each variable is mapped to its latent representation via a basic Multi-Layer Perceptron (MLP). We observe a significant degradation in performance. This substantial decline in accuracy underscores that the effectiveness of the Learnable Patch Replacement Module is highly contingent upon the quality of the underlying series embeddings. 3) *After removing the Token Filter module*, the model retains all tokens instead of filtering patch-wise tokens. We observe that on lower-quality datasets (such as ETTh2 and ETTm2), the performance drops significantly, highlighting the importance of token filtering in maintaining model accuracy, especially in time series data with noise or drift patterns. 4) *After removing the Feature Reduction module*, the token dimension is no longer compressed in the Replaced Flatten Head module. As a result, the model's performance generally declines, suggesting that feature compression not only reduces model complexity but also plays a role in enhancing overall predictive performance.

### 5.3.2. ROBUST STUDIES

To evaluate robustness, we conduct experiments under both synthetically perturbed conditions and real-world low-quality scenarios. Due to space constraints, we provide a brief description of the experimental setup below; more detailed experimental settings are available in Appendix C.1.

**Synthetically Perturbed Scenarios:** To evaluate the robustness of SEER under low-quality data conditions, we design a series of comparative experiments by injecting different types of perturbations into the ETTh2, including missing values, distribution shifts, anomalies, and white noise (see Table 5 and Figure 2). The prediction horizon is 96. The baseline models include SRSNet, Amplifier, iTransformer, and DLinear. Among them, SRSNet is inherently capable of handling non-stationarity, while Amplifier, iTransformer, and DLinear are enhanced with the RobustTSF (Cheng et al., 2024) plugin to improve their robustness. We can find that SEER consistently achieves the best performance under all four types of low-quality data conditions, demonstrating its superior robustness. Specifically, SEER significantly outperforms all baselines in the cases of missing values, anomalies, and white noise. In the distribution shift scenario, both SEER and SRSNet exhibit excellent results, indicating SEER's strong generalization capability when dealing with non-stationary time series.

**Real-world Low Quality Scenarios:** We select two representative datasets each for distribution shifts and white noise from the TFB (Qiu et al., 2024), along with two datasets each for anomalies from TAB (Qiu et al., 2025a) and missing values from APN (Liu et al., 2026b). For each dataset, we conduct evaluations across four different prediction horizons and report the average results in Table 6. The results demon-

*Table 6.* Average forecasting results under real-world low quality datasets. **Red**: the best, Blue: the 2nd best. Full results are available in Table 8 of Appendix C.1.

| Models | Distribution Shift | | | | White Noise | | | | Anomalies | | | | Missing Value | | | |
|---|---|---|---|---|---|---|---|---|---|---|---|---|---|---|---|---|
| Dataset | ILI | | NASDAQ | | FRED-MD | | Wike2000 | | SMD | | PSM | | USHCN | | Activity | |
| Metrics | mse | mae | mse | mae | mse | mae | mse | mae | mse | mae | mse | mae | mse | mae | mse | mae |
| DLinear | 2.729 | 1.130 | 1.370 | 0.839 | 77.426 | 1.512 | 669.115 | 1.417 | 0.562 | 0.393 | 1.287 | 0.757 | 0.767 | 0.504 | 0.017 | 0.119 |
| iTransformer | 2.054 | 0.910 | 1.065 | 0.729 | 69.145 | 1.391 | 540.908 | 1.169 | 0.655 | 0.380 | 1.286 | 0.744 | 0.772 | 0.505 | 0.018 | 0.107 |
| Amplifier | 2.238 | 0.927 | 1.085 | 0.734 | 73.791 | 1.478 | 561.262 | 1.160 | 0.578 | 0.351 | 1.291 | 0.783 | 0.761 | 0.495 | 0.019 | 0.106 |
| SRSNet | 2.049 | 0.884 | 1.030 | 0.711 | 61.245 | 1.308 | 506.372 | 1.111 | 0.603 | 0.357 | 1.276 | 0.741 | 0.758 | 0.494 | 0.019 | 0.104 |
| **SEER** | **1.876** | **0.829** | **0.953** | **0.686** | **54.266** | **1.273** | **474.559** | **1.061** | **0.520** | **0.293** | **1.147** | **0.693** | **0.754** | **0.492** | **0.016** | **0.103** |

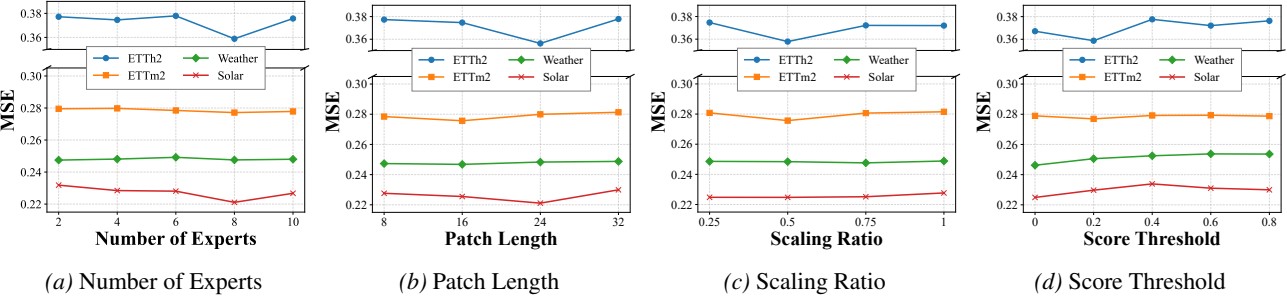

*(a)* Number of Experts     *(b)* Patch Length     *(c)* Scaling Ratio     *(d)* Score Threshold

*Figure 5.* Parameter sensitivity studies of main hyper-parameters in SEER.

strate that SEER maintains superior performance even in real-world low-quality scenarios, validating the effectiveness of our proposed Augmented Embedding Module and Learnable Patch Replacement Module. Overall, SRSNet ranks second only to SEER, while DLinear exhibits the weakest performance across the board.

### 5.3.3. PARAMETER SENSITIVITY

We also study the parameter sensitivity of the SEER. Figure 5a shows the results when the number of experts set in MOE changes. We observe that most datasets achieve their best performance when the number of experts is set to 8. Figure 5b shows the results when the patch length increases. We find that most datasets achieve optimal performance when the patch length is set to 16 or 24, while patch lengths that are too small or too large lead to performance degradation. Figure 5c shows the changes of scaling ratio in the Feature Reduction module. We find that the performance is best when the scaling ratio is 0.5 or 0.75. This also indicates that feature compression not only reduces the model's parameters, but also contributes to a certain degree of performance improvement. Finally, Figure 5d shows the results when the score threshold changes in the Automated Token Filter module. We find that most datasets perform best when the score threshold is non-zero, suggesting that discarding some negatively impactful tokens can enhance the model's performance. However, on certain high-quality datasets such as Solar, we observe that the optimal performance is achieved when the score threshold is set to zero.

## 6. Limitations

Despite its effectiveness, SEER has several limitations. First, in certain scenarios, abrupt changes may themselves represent critical signals (e.g., financial crashes or acute medical events) rather than noise. Although SEER learns filtering scores in an end-to-end manner, there remains a risk that such informative patterns could be mistakenly identified as low-quality patches and replaced. Second, the filtering threshold introduces sensitivity to hyperparameter selection; while it provides flexibility across datasets, it may also increase the tuning burden. Third, under severe global distribution shifts, the learned global prototypes may become less representative, potentially leading to biased predictions.

## 7. Conclusion

In this paper, we present SEER, a robust time series forecasting framework designed to address the inherent challenges of low-quality data in real-world scenarios. By moving beyond static patching techniques, SEER introduces an Augmented Embedding Module that leverages MoE and channel-adaptive mechanisms to refine patch and series-level representations. Furthermore, the Learnable Patch Replacement Module provides a dynamic defense against missing values, distribution shifts, and noise through its innovative two-stage filtering and substitution process. Extensive experiments across diverse datasets demonstrate that SEER not only achieves SOTA performance but also exhibits superior robustness compared to existing models.

## Acknowledgements

This work was partially supported by National Natural Science Foundation of China (62472174), the ECNU Multifunctional Platform for Innovation (001), and the Fundamental Research Funds for the Central Universities (YBNLTS2026010). Jilin Hu is the corresponding author of the work.

## Impact Statement

This paper presents work whose goal is to advance the field of Machine Learning. There are many potential societal consequences of our work, none which we feel must be specifically highlighted here.

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

# A. Robustness Experiment

To rigorously assess the robustness of our proposed SEER model, we conduct a series of controlled experiments that simulate real-world, low-quality data scenarios. We programmatically inject four distinct types of perturbations into the benchmark datasets: missing values, distribution shifts, anomalies, and white noise. These perturbations are designed to emulate common data imperfections encountered in practical applications, thereby enabling a comprehensive validation of the model's predictive resilience and accuracy under non-ideal conditions. The specific methodology for each perturbation is detailed in the subsequent algorithms.

## A.1. White Noise

To model high-frequency sensor jitter or random environmental interference as per **Definition 3.2**, we superimpose Gaussian noise onto a randomly selected subset of data points. The noise follows a zero-mean normal distribution with a standard deviation proportional to the channel's own standard deviation, assessing the model's stability against random, non-structural fluctuations. In our experiments, we set the noise ratio $r_{noise}$ to $\{0, 1\%, 5\%, 10\%, 15\%\}$ and fix the noise scale $\alpha_{noise}$ at 1.

---

**Algorithm 1** White Noise

---

**Input:** Time series $X \in \mathbb{R}^{T \times N}$, noise ratio $r_{noise}$, noise scale $\alpha_{noise}$.
**Output:** Noisy time series $X_{noisy} \in \mathbb{R}^{T \times N}$.

1: $X_{noisy} \leftarrow X$
2: $n_{noise} \leftarrow \lfloor T \cdot r_{noise} \rfloor$
3: **for** $c \in [1, N]$ **do**
4:  $\sigma_c \leftarrow \text{std}(X[:, c])$
5:  $I_{noise} \leftarrow \text{Sample}(\{0, ..., T-1\}, n_{noise})$ {Sample without replacement}
6:  Let $\epsilon$ be a vector of $n_{noise}$ i.i.d. samples from $\mathcal{N}(0, (\alpha_{noise} \cdot \sigma_c)^2)$
7:  $X_{noisy}[I_{noise}, c] \leftarrow X_{noisy}[I_{noise}, c] + \epsilon$
8: **end for**
9: **return** $X_{noisy}$

---

## A.2. Anomalies

We inject two forms of anomalies as described in **Definition 3.3**: 1) contiguous anomalous segments, simulating sustained system faults, and 2) discrete point outliers, simulating transient shocks. The anomaly magnitude is scaled by the channel's standard deviation. This composite approach provides a robust test of the model's capacity to identify and mitigate the influence of both prolonged and instantaneous abnormal events. In our experiments, we set the continuous anomaly ratio $r_{cont}$ to $\{0, 1\%, 5\%, 10\%, 15\%\}$, segment length $L_{cont}$ to 12, outlier ratio $r_{out}$ to 0.5%, and anomaly scale $\alpha_{anom}$ at 2.

---

**Algorithm 2** Anomalies

---

**Input:** Time series $X \in \mathbb{R}^{T \times N}$, continuous anomaly ratio $r_{cont}$, segment length $L_{cont}$, outlier ratio $r_{out}$, anomaly scale $\alpha_{anom}$.
**Output:** Noisy time series $X_{noisy} \in \mathbb{R}^{T \times N}$.

1: $X_{noisy} \leftarrow X$
2: {Inject continuous anomalous segments}
3: $n_{cont\_seg} \leftarrow \lfloor (T \cdot r_{cont})/L_{cont} \rfloor$
4: **for** $c \in [1, N]$ **do**
5:  $\sigma_c \leftarrow \text{std}(X[:, c])$
6:  $P_{cont} \leftarrow \{i \in \mathbb{Z} \mid 0 \le i \le T - L_{cont}\}$
7:  $S_{cont} \leftarrow \text{Sample}(P_{cont}, n_{cont\_seg})$
8:  **for** $s \in S_{cont}$ **do**
9:   $\delta_{anom} \leftarrow \text{Uniform}(\{-1, 1\}) \cdot \alpha_{anom} \cdot \sigma_c$
10:   $X_{noisy}[s : s + L_{cont}, c] \leftarrow X_{noisy}[s : s + L_{cont}, c] + \delta_{anom}$
11:  **end for**
12: **end for**
13: {Inject discrete point outliers}
14: $n_{out} \leftarrow \lfloor T \cdot r_{out} \rfloor$
15: **for** $c \in [1, N]$ **do**
16:  $\sigma_c \leftarrow \text{std}(X[:, c])$
17:  $I_{out} \leftarrow \text{Sample}(\{0, ..., T-1\}, n_{out})$
18:  **for** $i \in I_{out}$ **do**
19:   $\delta_{out} \leftarrow \text{Uniform}(\{-1, 1\}) \cdot \alpha_{anom} \cdot \sigma_c$
20:   $X_{noisy}[i, c] \leftarrow X_{noisy}[i, c] + \delta_{out}$
21:  **end for**
22: **end for**
23: **return** $X_{noisy}$

---

## A.3. Missing Values

To simulate contiguous data loss, such as periods of sensor malfunction or network outage, we introduce segments of zero values into the time series. This method directly tests the model's ability to handle incomplete information and impute missing semantics, corresponding to the scenario outlined in **Definition 3.4**. In our experiments, we set the missing ratio $r_{miss}$ to $\{0, 1\%, 5\%, 10\%, 15\%\}$ and segment length $L_{miss}$ at 12.

---

**Algorithm 3** Missing Values

---

**Input:** Time series $X \in \mathbb{R}^{T \times N}$, missing ratio $r_{miss}$, segment length $L_{miss}$.
**Output:** Noisy time series $X_{noisy} \in \mathbb{R}^{T \times N}$.

1: $X_{noisy} \leftarrow X$
2: $n_{seg} \leftarrow \lfloor (T \cdot r_{miss})/L_{miss} \rfloor$
3: **for** $c \in [1, N]$ **do**
4:  $P \leftarrow \{i \in \mathbb{Z} \mid 0 \le i \le T - L_{miss}\}$
5:  $S \leftarrow \text{Sample}(P, n_{seg})$
6:  **for** $s \in S$ **do**
7:   $X_{noisy}[s : s + L_{miss}, c] \leftarrow \mathbf{0}$
8:  **end for**
9: **end for**
10: **return** $X_{noisy}$

---

**Algorithm 4** Distribution Shift

---

**Input:** Time series $X \in \mathbb{R}^{T \times N}$, number of segments $K_{shift}$, shift scale $\alpha_{shift}$.
**Output:** Noisy time series $X_{noisy} \in \mathbb{R}^{T \times N}$.

1: $X_{noisy} \leftarrow X$
2: $L_{seg} \leftarrow \lfloor T/K_{shift} \rfloor$
3: **for** $c \in [1, N]$ **do**
4:    $\sigma_c \leftarrow \text{std}(X[:, c])$
5:    **for** $k \in [0, K_{shift} - 1]$ **do**
6:       start $\leftarrow k \cdot L_{seg}$
7:       end $\leftarrow \min((k + 1) \cdot L_{seg}, T)$
8:       $\delta \leftarrow \text{Uniform}(-\alpha_{shift}, \alpha_{shift}) \cdot \sigma_c$
9:       $X_{noisy}[\text{start} : \text{end}, c] \leftarrow X_{noisy}[\text{start} : \text{end}, c] + \delta$
10:    **end for**
11: **end for**
12: **return** $X_{noisy}$

---

## A.4. Distribution Shift

To emulate the non-stationarity inherent in many real-world processes (i.e., concept drift, as per **Definition 3.5**), we partition the time series and apply a distinct offset to each segment. The magnitude of the shift is scaled by the standard deviation of each channel, ensuring the perturbation is contextually relevant to the series' intrinsic volatility. This experiment evaluates the model's adaptability to evolving data distributions. In our experiments, we set the number of segments $K_{shift}$ to $\{0, 1, 3, 5, 10\}$ and the shift scale $\alpha_{shift}$ at 5.

# B. Other Related Works

## B.1. Time Series Forecasting Methods

Time series forecasting plays a crucial role in a wide range of domains (Shang et al., 2026; 2024; Chen et al., 2023; Shang & Chen, 2024). Traditional time series forecasting methods, such as ARIMA (Box & Pierce, 1970), ETS (Hyndman et al., 2008), Theta (Federico Garza, 2022), VAR (Godahewa et al., 2021), Random Forests (Breiman, 2001), and LightGBM (Ke et al., 2017), have been widely studied and applied in practical forecasting tasks. However, these methods generally rely on manually designed features and predefined statistical assumptions, which limit their flexibility and scalability when dealing with complex temporal patterns (Fang et al., 2025; Li et al., 2026a; Zhang et al., 2024; 2025b; Ni et al., 2026; Chen et al., 2025; Qi et al., 2025; Ma et al., 2024; 2025b; 2022; Lu et al., 2024b). With the rapid advancement of deep learning, neural forecasting models have demonstrated remarkable capabilities in automatically learning temporal representations from raw historical data (Fang et al., 2023; 2022; Li et al., 2026b; Zhang et al., 2025a;c; Wang et al., 2026d; Ma et al., 2026; 2025a; Yu et al., 2025d;c;e; Lu et al., 2024a; 2023). Numerous neural forecasting models have emerged in recent years (Yang et al., 2026; 2025b; Hu et al., 2026; Yang et al., 2025a). Transformer-based architectures, such as Informer (Zhou et al., 2021), FEDformer (Zhou et al., 2022), Autoformer (Wu et al., 2021), Triformer (Cirstea et al., 2022), and PatchTST (Nie et al., 2023), have achieved strong performance by effectively modeling long-range temporal dependencies and complex token interactions. Meanwhile, MLP-based approaches, including SparseTSF (Lin et al., 2024a), CycleNet (Lin et al., 2024b), DUET (Qiu et al., 2025c), NLinear (Zeng et al., 2023), and DLinear (Zeng et al., 2023), demonstrate that lightweight architectures with fewer parameters can still achieve highly competitive forecasting accuracy.

# C. Full Results

## C.1. Robust Time Series Forecasting

To evaluate robustness, we conduct experiments under both synthetically perturbed conditions and real-world low-quality scenarios.

**Synthetically Perturbed Scenarios:** To evaluate the robustness of SEER under low-quality data conditions, we design a series of comparative experiments by injecting different types of perturbations into the ETTh2, including missing values, distribution shifts, anomalies, and white noise (see Table 5 and Figure 2). The prediction horizon is 96. The baseline models include SRSNet, Amplifier, iTransformer, and DLinear. Among them, SRSNet is inherently capable of handling non-stationarity, while Amplifier, iTransformer, and DLinear are enhanced with the RobustTSF (Cheng et al., 2024) plugin to improve their robustness. The injection settings for the four types of perturbations—missing values, distribution shifts, anomalies, and white noise—are detailed in Appendix A. We can find that SEER consistently achieves the best performance under all four types of low-quality data conditions, demonstrating its superior robustness. Specifically, SEER significantly outperforms all baselines in the cases of missing values, anomalies, and white noise. In the distribution shift scenario, both SEER and SRSNet exhibit excellent results, indicating SEER's strong generalization capability when dealing with non-stationary time series.

**Real-world Low Quality Scenarios:** We select two representative datasets for each type of data degradation.

*For distribution shifts and white noise*, datasets are selected from TFB (Qiu et al., 2024). Specifically, ILI and NASDAQ are chosen as the most distribution-shifted datasets according to the shifting metric defined in TFB, while FRED-MD and Wiki2000 are selected as the datasets with the highest degree of white noise based on the Ljung–Box statistical test (Box et al., 2015).

*For anomalies*, we adopt two datasets from TAB (Qiu et al.,

*Table 7.* Robustness comparison of different models under various synthetically perturbed scenarios.

| Models | Metrics | ori | White Noise | | | | | Anomalies | | | | | Missing Values | | | | | Distribution Shift | | | | |
|---|---|---|---|---|---|---|---|---|---|---|---|---|---|---|---|---|---|---|---|---|---|---|
| | | | 1% | 5% | 10% | 15% | avg | 1% | 5% | 10% | 15% | avg | 1% | 5% | 10% | 15% | avg | 1 | 3 | 5 | 10 | avg |
| DLinear | mse | 0.318 | 0.319 | 0.319 | 0.320 | 0.313 | 0.318 | 0.311 | 0.309 | 0.369 | 0.386 | 0.344 | 0.318 | 0.325 | 0.332 | 0.327 | 0.326 | **0.288** | 0.305 | 0.322 | 0.325 | 0.310 |
| | mae | 0.374 | 0.375 | 0.375 | 0.377 | 0.369 | 0.374 | 0.368 | 0.366 | 0.418 | 0.430 | 0.396 | 0.369 | 0.382 | 0.381 | 0.376 | 0.377 | 0.339 | 0.354 | 0.379 | 0.383 | 0.364 |
| iTransformer | mse | 0.304 | 0.315 | 0.314 | 0.310 | 0.302 | 0.310 | 0.305 | 0.309 | 0.319 | 0.312 | 0.311 | 0.306 | 0.320 | 0.308 | 0.307 | 0.310 | 0.302 | 0.318 | 0.324 | 0.319 | 0.316 |
| | mae | 0.353 | 0.362 | 0.354 | 0.354 | 0.349 | 0.355 | 0.353 | 0.353 | 0.358 | 0.358 | 0.356 | 0.354 | 0.364 | 0.354 | 0.353 | 0.356 | 0.353 | 0.362 | 0.365 | 0.359 | 0.360 |
| Amplifier | mse | 0.308 | 0.310 | 0.300 | 0.303 | 0.295 | 0.302 | 0.307 | 0.311 | **0.295** | 0.311 | 0.306 | 0.305 | 0.300 | 0.309 | 0.314 | 0.307 | 0.306 | 0.310 | 0.306 | 0.303 | 0.306 |
| | mae | 0.359 | 0.359 | 0.350 | 0.353 | 0.344 | 0.352 | 0.358 | 0.356 | 0.344 | 0.357 | 0.356 | 0.356 | 0.352 | 0.356 | 0.361 | 0.356 | 0.357 | 0.353 | 0.352 | 0.351 | 0.353 |
| SRSNet | mse | 0.296 | 0.298 | 0.296 | 0.296 | 0.299 | 0.297 | 0.295 | **0.301** | 0.306 | 0.307 | 0.302 | 0.301 | 0.302 | 0.296 | 0.297 | 0.299 | 0.295 | 0.301 | 0.308 | 0.306 | 0.303 |
| | mae | 0.345 | 0.346 | 0.345 | 0.345 | 0.348 | 0.346 | 0.345 | 0.350 | 0.354 | 0.355 | 0.351 | 0.349 | 0.350 | 0.345 | 0.346 | 0.348 | 0.342 | 0.346 | 0.348 | 0.350 | 0.347 |
| **SEER** | mse | **0.281** | **0.298** | **0.294** | **0.291** | **0.289** | **0.293** | **0.287** | 0.310 | **0.303** | **0.306** | **0.302** | **0.291** | **0.289** | **0.291** | **0.289** | **0.290** | 0.289 | **0.299** | **0.295** | **0.301** | **0.296** |
| | mae | **0.329** | **0.340** | **0.336** | **0.334** | **0.334** | **0.336** | **0.332** | **0.340** | **0.337** | **0.341** | **0.338** | **0.336** | **0.334** | **0.334** | **0.334** | **0.335** | **0.332** | **0.344** | **0.336** | **0.348** | **0.340** |

2025a): SMD, which mainly contains point anomalies with an anomaly ratio of 1.48% in the training set, and PSM, which primarily includes subsequence anomalies with a training anomaly ratio of 14.2%. Both datasets are truncated to ensure that the test set is free of anomalies.

*For missing values*, we use two datasets from PyOmniTS (Li et al., 2025a) and APN (Liu et al., 2026b): USHCN, with a missing rate of 77.9%, and Human Activity, with a missing rate of 75%. To ensure strict fairness, we rely on the standardized preprocessing pipeline natively provided by the PyOmniTS framework. While HyperIMTS is a GNN-based model, its official PyOmniTS framework was explicitly built as a unified, extensible pipeline to benchmark both irregular-specific models and standard regular-time models (e.g., MLPs, Transformers) on a level playing field. Specifically, for SEER and all regular-time baselines, the framework handles the irregular-to-regular adaptation uniformly through the following steps:

1) Conversion into Model-Compatible Inputs (Canonical Pre-Alignment): Irregular and unaligned observations across different variables are mapped onto a unified global temporal grid (typically the union of all unique timestamps within a given window). To construct the standard dense tensor format required by regular MLPs/Transformers, missing values at these aligned timestamps are padded with zeros.

2) Representation of Timestamps and Time Gaps: To prevent the model from treating padded zeros as actual sensor readings, the framework simultaneously generates a corresponding binary observation mask (1 for observed, 0 for padded). This mask, alongside explicit temporal features (such as time-gap intervals or continuous timestamp embeddings), is appended/concatenated to the input. This ensures that regular models are explicitly aware of the irregular sampling intervals and the exact locations of the missing values.

3) Absolute Consistency Across Baselines: We emphasize that this exact same adaptation process—including the grid alignment, zero-padding, masking, and timestamp representation—is applied strictly and consistently to SEER and all regular-time baselines. They all receive identical pre-aligned tensors from the PyOmniTS data loaders.

By utilizing this unified preprocessing wrapper, we ensure that the performance differences observed in our benchmark are purely attributable to the architectural designs of the models themselves, rather than any discrepancies in data handling.

For each dataset, we conduct evaluations across four different prediction horizons and report the average results in Table 6. The results demonstrate that SEER maintains superior performance even in real-world low-quality scenarios, validating the effectiveness of our proposed Augmented Embedding Module and Learnable Patch Replacement Module. SEER also consistently outperforms SRSNet, indicating its stronger performance on robust forecasting. Overall, SRSNet ranks second only to SEER, while DLinear exhibits the weakest performance across the board.

*Table 8.* Full forecasting results under real-world low quality datasets. **Red**: the best, Blue: the 2nd best.

| | Models | | SEER (ours) | | SRSNet (2025) | | Amplifier (2025) | | iTransformer (2024) | | DLinear (2023) | |
|---|---|---|---|---|---|---|---|---|---|---|---|---|
| | Metrics | | mse | mae | mse | mae | mse | mae | mse | mae | mse | mae |
| Distribution Shift | ILI | 24 | 1.872 | 0.817 | 2.008 | 0.860 | 2.374 | 0.940 | 1.908 | 0.872 | 2.684 | 1.112 |
| | | 36 | 1.911 | 0.836 | 2.121 | 0.872 | 2.220 | 0.926 | 2.023 | 0.892 | 2.715 | 1.134 |
| | | 48 | 1.877 | 0.827 | 2.108 | 0.896 | 2.219 | 0.923 | 2.085 | 0.909 | 2.667 | 1.113 |
| | | 60 | 1.844 | 0.836 | 1.957 | 0.908 | 2.137 | 0.918 | 2.201 | 0.968 | 2.849 | 1.162 |
| | | avg | 1.876 | 0.829 | 2.049 | 0.884 | 2.238 | 0.927 | 2.054 | 0.910 | 2.729 | 1.130 |
| | NASDAQ | 24 | 0.530 | 0.510 | 0.572 | 0.530 | 0.649 | 0.564 | 0.617 | 0.546 | 0.809 | 0.647 |
| | | 36 | 0.806 | 0.643 | 0.914 | 0.667 | 0.950 | 0.705 | 0.993 | 0.734 | 1.240 | 0.796 |
| | | 48 | 1.136 | 0.769 | 1.225 | 0.794 | 1.353 | 0.827 | 1.327 | 0.814 | 1.773 | 0.979 |
| | | 60 | 1.340 | 0.820 | 1.409 | 0.853 | 1.388 | 0.841 | 1.325 | 0.820 | 1.659 | 0.935 |
| | | avg | 0.953 | 0.686 | 1.030 | 0.711 | 1.085 | 0.734 | 1.065 | 0.729 | 1.370 | 0.839 |
| White Noise | FRED-MD | 24 | 23.808 | 0.853 | 28.499 | 0.893 | 35.931 | 1.048 | 26.832 | 0.896 | 37.898 | 1.070 |
| | | 36 | 41.298 | 1.139 | 48.765 | 1.181 | 59.960 | 1.349 | 46.744 | 1.215 | 58.262 | 1.352 |
| | | 48 | 63.216 | 1.422 | 78.054 | 1.513 | 80.834 | 1.609 | 78.494 | 1.542 | 96.457 | 1.706 |
| | | 60 | 88.741 | 1.676 | 89.661 | 1.646 | 118.439 | 1.906 | 124.511 | 1.913 | 117.086 | 1.919 |
| | | avg | 54.266 | 1.273 | 61.245 | 1.308 | 73.791 | 1.478 | 69.145 | 1.391 | 77.426 | 1.512 |
| | Wike2000 | 24 | 415.872 | 0.927 | 457.232 | 1.003 | 514.766 | 1.064 | 467.080 | 1.028 | 592.252 | 1.307 |
| | | 36 | 463.738 | 1.032 | 504.389 | 1.095 | 529.391 | 1.127 | 528.797 | 1.147 | 651.401 | 1.380 |
| | | 48 | 487.536 | 1.095 | 533.838 | 1.170 | 581.429 | 1.192 | 574.776 | 1.229 | 694.854 | 1.454 |
| | | 60 | 531.088 | 1.190 | 530.029 | 1.177 | 619.464 | 1.258 | 592.980 | 1.273 | 737.955 | 1.526 |
| | | avg | 474.559 | 1.061 | 506.372 | 1.111 | 561.262 | 1.160 | 540.908 | 1.169 | 669.115 | 1.417 |
| Anomalies | SMD | 96 | 0.338 | 0.201 | 0.391 | 0.258 | 0.439 | 0.215 | 0.445 | 0.283 | 0.460 | 0.327 |
| | | 192 | 0.420 | 0.249 | 0.482 | 0.306 | 0.464 | 0.308 | 0.534 | 0.329 | 0.552 | 0.391 |
| | | 336 | 0.548 | 0.309 | 0.633 | 0.374 | 0.616 | 0.346 | 0.689 | 0.396 | 0.609 | 0.405 |
| | | 720 | 0.772 | 0.413 | 0.905 | 0.491 | 0.792 | 0.536 | 0.950 | 0.511 | 0.627 | 0.447 |
| | | avg | 0.520 | 0.293 | 0.603 | 0.357 | 0.578 | 0.351 | 0.655 | 0.380 | 0.562 | 0.393 |
| | PSM | 96 | 0.381 | 0.399 | 0.421 | 0.428 | 0.415 | 0.428 | 0.418 | 0.423 | 0.419 | 0.469 |
| | | 192 | 0.725 | 0.549 | 0.789 | 0.590 | 0.791 | 0.617 | 0.820 | 0.602 | 0.781 | 0.594 |
| | | 336 | 1.280 | 0.750 | 1.364 | 0.801 | 1.315 | 0.858 | 1.454 | 0.823 | 1.377 | 0.811 |
| | | 720 | 2.203 | 1.072 | 2.531 | 1.146 | 2.643 | 1.227 | 2.452 | 1.130 | 2.572 | 1.153 |
| | | avg | 1.147 | 0.693 | 1.276 | 0.741 | 1.291 | 0.783 | 1.286 | 0.744 | 1.287 | 0.757 |
| Missing Value | USHCN | 1 | 0.762 | 0.475 | 0.764 | 0.483 | 0.777 | 0.483 | 0.823 | 0.517 | 0.800 | 0.499 |
| | | 6 | 0.744 | 0.501 | 0.746 | 0.495 | 0.745 | 0.493 | 0.743 | 0.495 | 0.745 | 0.496 |
| | | 12 | 0.751 | 0.486 | 0.751 | 0.495 | 0.752 | 0.496 | 0.751 | 0.497 | 0.752 | 0.508 |
| | | 24 | 0.760 | 0.506 | 0.770 | 0.504 | 0.770 | 0.506 | 0.772 | 0.512 | 0.771 | 0.509 |
| | | avg | 0.754 | 0.492 | 0.758 | 0.494 | 0.761 | 0.495 | 0.772 | 0.505 | 0.767 | 0.504 |
| | Activity | 200 | 0.016 | 0.101 | 0.019 | 0.104 | 0.018 | 0.104 | 0.016 | 0.100 | 0.016 | 0.102 |
| | | 1200 | 0.016 | 0.102 | 0.019 | 0.105 | 0.018 | 0.107 | 0.015 | 0.097 | 0.017 | 0.104 |
| | | 3000 | 0.017 | 0.106 | 0.018 | 0.104 | 0.019 | 0.106 | 0.015 | 0.097 | 0.017 | 0.107 |
| | | 6000 | 0.016 | 0.104 | 0.019 | 0.104 | 0.019 | 0.108 | 0.025 | 0.133 | 0.018 | 0.162 |
| | | avg | 0.016 | 0.103 | 0.019 | 0.104 | 0.019 | 0.106 | 0.018 | 0.107 | 0.017 | 0.119 |

