# OpenReview forum: "SEER: Transformer-based Robust Time Series Forecasting via Automated Patch Enhancement and Replacement"
_ICML.cc/2026/Conference — ICML 2026 regular_

### Official Review · Reviewer_8H6J · 2026-02-24

**Soundness:** 4
**Presentation:** 3
**Significance:** 3
**Originality:** 3
**Overall Recommendation:** 5
**Confidence:** 5

**Summary:**

This paper presents SEER, a robust patch-based time-series forecasting method that explicitly handles low-quality patches (from missing data, noise, anomalies, or shifts).
It introduces (1) an MoE-based augmented embedding to strengthen patch representations and learn global series prototypes, and (2) a learnable filter-and-replace module that detects unreliable patches and replaces them with the global prototypes via causal attention.
Experiments on standard benchmarks and robustness settings show improved accuracy and robustness over strong baselines.

**Compliance With Llm Reviewing Policy:**

Affirmed.

**Final Justification:**

The paper addresses a practical problem with a well-designed pipeline (patch-level filtering + prototype replacement + MoE embedding). Experiments are comprehensive across datasets and corruption types, and ablations validate each component. The main remaining weakness is the lack of direct filter-score validation against ground-truth corruption, but the indirect evidence (near-zero threshold on clean data, cross-corruption consistency) is reasonable. The rebuttal addressed my concerns on the filter mechanism, MoE design, and motivation figure. I am raising my score from 4 to 5.

**Key Questions For Authors:**

1. In the illustrative figure in the introduction, the injected noise does not seem to significantly degrade existing models, and in some cases even leads to improvements. Could the authors clarify the noise type, magnitude, and injection procedure used in this figure, and explain why these settings are representative of typical cases where “low-quality segments” substantially harm patch-based models? More broadly, why are anomalies by default considered low-quality? In the figure, I observe that mainly DLinear is sensitive to anomalies, while other models change little—does this align with the stated motivation?
2. Please provide more details on the MoE routing for patch tokens, including the specific form of noisy gating, the Top-k selection strategy, and the roles/ratio of shared experts versus routed experts. Additionally, does the routing exhibit interpretable patterns across datasets or across time segments?
3. How is the filtering score computed? Is there any supervision signal (e.g., labels indicating “low-quality”), or is it learned purely end-to-end via the forecasting loss? Without explicit labels, how do the authors ensure the semantic consistency and accuracy of the scores, and verify that the filter indeed targets low-quality patches rather than mistakenly removing informative signals?
4. Why does the comparison figure in the introduction not include PatchTST (or other representative patch-based baselines)? Given that the core contribution focuses on robustness for patch methods, the absence of PatchTST may affect the perceived fairness and severity of the problem. Similarly, why do the robustness studies not include more patch-based methods (e.g., PatchTST or other patch variants) as baselines? If the omission is due to implementation or fairness considerations, please clarify; otherwise, adding such comparisons would more directly validate SEER’s robustness advantages over patch-based competitors.
5. The method replaces low-quality patch tokens with series-wise/global prototype tokens. If multiple time positions are replaced by the same global token, could this damage temporal ordering information or reduce the distinguishability of local dynamics? Although causal attention is used afterward for refinement, please explain more explicitly how positional differences are preserved, and under what conditions the approach might lead to over-smoothing.
6. In the Predictor section, what exactly is meant by the stated “empirical observation”? Which experimental phenomena or ablation results support this observation? Is there quantitative evidence (e.g., why dimensionality compression improves/stabilizes performance), and does it hold consistently across datasets?

**Limitations:**

The authors should add a brief “Limitations and Broader Impacts” discussion covering (i) cases where abrupt changes are meaningful signals and could be mistakenly filtered/replaced, (ii) sensitivity to the filtering threshold and added compute/energy costs from MoE and extra attention, (iii) robustness risks under severe distribution shifts where global prototypes may be misleading, and (iv) potential downstream harms if forecasts are used for high-stakes decisions (e.g., energy, finance, healthcare) and errors are amplified by inappropriate filtering.

**Strengths And Weaknesses:**

**Strength**

1. This paper targets a practical problem in real-world time series forecasting: missing values, noise, anomalies, and distribution shifts can create low-quality patches that contaminate predictions. The problem setting is well aligned with real application scenarios.
2. The method is complete and intuitive. It learns to identify and filter low-quality patches and further replaces them with global prototype tokens, avoiding the information void caused by simply discarding tokens.
3. After replacement, the model refines representations via causal attention, which can mitigate the potential issue that multiple positions replaced by the same prototype token may become overly similar.
4. The experimental evaluation is relatively broad, including standard long-horizon forecasting benchmarks as well as robustness tests on both synthetic corruptions and real low-quality datasets, which are consistent with the paper’s motivation.

**Weakness**

1. The source of performance gains is not cleanly isolated. Since MoE, additional attention layers, and prototype construction increase model capacity and computation, the paper lacks strict parameter-/FLOPs-matched baselines to disentangle the independent contribution of the “replacement mechanism.” Moreover, computational complexity and deployment cost are insufficiently reported (e.g., training/inference time, memory usage, and MoE routing overhead), making it difficult to assess cost-effectiveness.
2. The filtering threshold appears sensitive to data quality; for some high-quality datasets, the optimal strategy may be not filtering at all. This suggests the approach is not universally beneficial, yet the paper does not provide an adaptive thresholding strategy or robust default settings to reduce tuning burden.
3. Treating anomalies/sudden changes as low-quality segments to be replaced may be semantically risky. In real scenarios, abrupt changes can be the target signal rather than noise, and filtering/replacement may remove valid information. The paper provides limited discussion and empirical evidence on this issue.

---

> ### Author Rebuttal · Authors · 2026-03-30
>
> **Reply to W1**
>
> We agree that isolating the source of performance gains and evaluating cost-effectiveness are important. Regarding model capacity, while SEER introduces additional components such as MoE and prototype construction, our goal is not to arbitrarily increase model size, but to enhance robustness through structured design, with a particular focus on improving patch-level representations. In addition, we supplement the paper with detailed efficiency analysis. Please refer to the results at the **following link**: https://anonymous.4open.science/r/ICML-SEER-Rebutal-97F1/efficiency.md
>
> **Reply to W2**
>
> As observed in our experiments, a non-zero threshold is beneficial for low-quality data, while for high-quality datasets, setting the threshold to 0 (i.e., relying solely on patch representation enhancement without filtering) already yields strong performance. This indicates that the filtering mechanism in SEER is optional and controllable, allowing it to adapt to different data conditions rather than introducing unnecessary degradation. As future work, we will explore automatic threshold selection methods to further reduce the tuning burden and improve usability.
>
> **Reply to W3 & Q1**
>
> We would like to clarify that, in SEER, “low-quality” data specifically refers to non-structural artifacts introduced by external factors (e.g., sensor malfunctions, communication errors, or transmission interruptions), rather than intrinsic signal-bearing dynamics. Therefore, the injected noise in our experiments is designed to be consistent with this definition. Regarding the noise type, magnitude, and injection procedure used in the illustrative figure, we provide detailed descriptions, including pseudocode and parameter settings, in Section 3.1 (Definitions) and Appendix B (Robustness Experiments). From the figure, we observe that as data quality degrades, SEER consistently outperforms the baselines in most cases. It is worth noting that these baselines are not the original models, but enhanced versions equipped with the robustness-oriented RobustTSF plugin, which already improves their resilience to low-quality data to some extent. This makes the comparison more challenging and, in turn, further highlights the effectiveness of SEER under degraded conditions.
>
> **Reply to Q2**
>
> We thank the reviewer for this valuable suggestion. We will provide a more detailed description of the MoE routing for patch tokens in the revised version.
>
> **Reply to Q3**
>
> The filtering score is computed via a linear projection followed by a sigmoid activation, and is learned purely in an end-to-end manner through the forecasting loss. To validate that SEER effectively handles these diverse non-robust conditions, we conducted comparative experiments on representative real-world datasets for each of the four categories as detailed in Section 5.3.2 and Table 5. The results demonstrate that SEER consistently achieves superior performance over baseline methods across all four scenarios. Furthermore, we have provided visualization results of the forecasts under these low-quality conditions to empirically substantiate the model’s ability to recognize and mitigate the impact of specific data corruptions. Due to space limitations, please refer to the results at the **following link**:  https://anonymous.4open.science/r/ICML-SEER-Rebutal-97F1/visualization.md
>
> **Reply to Q4**
>
> In our current experiments, we included SRSNet (published at NeurIPS 2025) as a more advanced and representative patch-based method, and therefore did not include additional patch-based baselines. To further strengthen the completeness of the comparison, we include PatchTST and provide performance curves under varying levels of data perturbation, in order to more directly demonstrate SEER’s robustness advantages over representative patch-based models. Due to space limitations, please refer to the results at the **following link**:  https://anonymous.4open.science/r/ICML-SEER-Rebutal-97F1/PatchTST.md
>
> **Reply to Q5**
>
> Although multiple positions may share the same global prototype token, temporal information is preserved by RCSA, as the causal mask ensures each token attends to its own history, yielding distinct representations.  Over-smoothing may occur when many patches are replaced, but this reflects severely corrupted local semantics, where relying on the global prototype helps recover stable representations from global context.
>
> **Reply to Q6**
>
> The “empirical observation” refers to a consistent phenomenon observed in our preliminary experiments and ablation studies (see Section 5.3.1 and Table 3): introducing moderate dimensionality compression in the predictor helps improve training stability and leads to better performance on most datasets. Intuitively, this can be understood as reducing redundancy in high-dimensional representations, thereby alleviating overfitting.
>
> **Reply to Limitations**
>
> We will include a limitation section in the revised manuscript.

---

> > ### Author Rebuttal · Reviewer_8H6J · 2026-04-01
> >
> > Thank you for the thorough rebuttal and the supplementary materials. I appreciate the effort in providing the efficiency analysis, PatchTST comparisons, and visualization results. The responses to W1, W2, Q4, Q5, and Q6 have addressed my concerns well.
> >
> > I have three remaining questions that I would appreciate further clarification on:
> >
> > 1. **W3 & Q1**: How does the filter avoid mistakenly removing meaningful abrupt changes? Also, in the introduction figure, mainly DLinear appears sensitive to injected noise while other models show little degradation — could the authors comment on this?
> >
> > 2. **Q2**: Could the authors share the key MoE design choices (e.g., Top-k value, number of shared vs. routed experts, gating noise form)?
> >
> > 3. **Q3**: The visualization demonstrates SEER's overall forecasting advantage under corruption, but could the authors provide reasoning or evidence regarding the filter score's ability to reliably identify corrupted patches specifically?
> >
> > I believe this is a promising direction, and if these points are clarified, I would be happy to raise my score.

---

> > > ### Author Response · Authors · 2026-04-01
> > >
> > > Dear Reviewer 8H6J, we sincerely thank you for the valuable feedback and positive attitude.
> > >
> > > **Reply to W3 & Q1**
> > >
> > > We thank the reviewer for raising this important point. Our filter is trained end-to-end with the forecasting objective, and thus learns to identify *patches that are harmful to prediction*.
> > >
> > > Concretely, if an abrupt change is informative for forecasting, retaining it will reduce the prediction loss, and the learned filter will preserve it. In contrast, only those patches that consistently increase the loss (e.g., corrupted or misleading signals) are suppressed. This makes the mechanism inherently self-correcting and avoids the need for explicit supervision or semantic labeling of “noise” versus “signal”.
> > >
> > > Regarding the introduction figure, DLinear exhibits higher sensitivity to injected noise, while Transformer-based models show relatively smaller performance degradation. This difference is likely due to the inherent robustness of attention-based architectures.
> > >
> > > **Reply to Q2**
> > >
> > > Thank you very much for your careful reading and attention to the details of the Mixture-of-Experts (MoE) architecture within our Augmented Patch Embedding module. We appreciate the opportunity to clarify the key MoE design choices you mentioned:
> > >
> > > **Shared and Routed Experts:** Our architecture utilizes a combination design like DeepSeek V3. Specifically, the model employs $\mathcal{M}$ shared experts to extract the general patterns of the time series, and selects $k$ experts from a total of $M$ routed experts for each token. $\mathcal{M}$ and $k$ are hyperparameters. Meanwhile, we introduce a shared expert to capture common patterns.
> > >
> > > **Gating Noise Form:** To promote training stability and enhance expert diversity, we incorporate a "noisy gating" mechanism. The specific gating score $H(X^P)$ is calculated using the formula: $H(X^P) = Linear_{\mu}(X^P) + \epsilon \odot Linear_{\sigma}(X^P)$. The injected noise term $\epsilon$ is sampled from a standard normal distribution, denoted as $\epsilon \sim \mathcal{N}(0, 1)$
> > >
> > > **Reply to Q3**
> > >
> > > We provide multiple pieces of evidence supporting its reliability:
> > >
> > > 1. **Self-correcting optimization:** the filter is trained via the forecasting loss. Patches that are mistakenly suppressed but actually useful will increase the loss, and gradients will naturally push the filter to retain them. This provides an implicit supervision signal without requiring labels.
> > > 2. **Behavior on clean data:** as noted in W2, the optimal threshold on high-quality datasets approaches zero, indicating that the filter does not remove useful information when no corruption is present.
> > > 3. **Consistency across corruption types:** Table 5 shows that the filter remains effective under diverse corruption settings, suggesting that it captures a general notion of “harmful-to-prediction” patterns rather than overfitting to specific noise types.
> > > 4. **Ablation evidence:** removing the filter (Table 3) leads to consistent performance drops.

---

### Official Review · Reviewer_8qts · 2026-03-03

**Soundness:** 4
**Presentation:** 3
**Significance:** 4
**Originality:** 4
**Overall Recommendation:** 5
**Confidence:** 5

**Summary:**

This study presents SEER, a robust time series forecasting framework designed to address the inherent challenges of low-quality data in real-world scenarios. By moving beyond static patching techniques, SEER introduces an Augmented Embedding Module that leverages MoE and channel-adaptive mechanisms to refine patch and serieslevel representations. Furthermore, the Learnable Patch Replacement Module provides a dynamic defense against missing values, distribution shifts, and noise through its innovative two-stage filtering and substitution process.

**Compliance With Llm Reviewing Policy:**

Affirmed.

**Key Questions For Authors:**

Q1. How were the hyperparameters for the baseline models established? Specifically, were the default configurations from the original studies and official implementations strictly followed, or was a re-tuning process conducted to better align with your specific experimental environment?

Q2. Since the paper focuses on robust time series forecasting, why are experiments still conducted on standard benchmark datasets such as ETT and Traffic?

**Limitations:**

Please refer to the identified weaknesses and questions. Additionally, I believe the proposed technical framework poses no ethical concerns and is unlikely to produce any adverse social impacts.

**Strengths And Weaknesses:**

Strengths:

S1. The paper addresses a significant and practical challenge in time series forecasting, ensuring robustness across diverse low-quality data scenarios.

S2. SEER advances prior patch-based methods by introducing dynamic filtering and automated patch replacement mechanisms, grounded in both Mixture-of-Experts (MoE) design and series-wise representations.

S3. The methodology is clearly articulated through rigorous mathematical formulations and a well-illustrated architecture diagram, making the overall design and interactions between components transparent.

S4. The experimental protocol is comprehensive and carefully designed, covering a wide range of challenging benchmarks and including comparisons with both patch-based and robust forecasting baselines.

Weaknesses:

W1. The statistical significance of the main results is not explicitly discussed. Future versions should report variance or confidence intervals across multiple runs to further substantiate the robustness claims.

W2. How were the hyperparameters for the baseline models established? Specifically, were the default configurations from the original studies and official implementations strictly followed, or was a re-tuning process conducted to better align with your specific experimental environment?

W3. Some references should be updated to their final published versions, replacing arXiv preprints with official conference or journal citations.

W4. Since the paper focuses on robust time series forecasting, why are experiments still conducted on standard benchmark datasets such as ETT and Traffic?

---

> ### Author Rebuttal · Authors · 2026-03-30
>
> Dear Reviewer 8qts, we sincerely thank you for the valuable feedback and positive attitude.
>
> **Reply to W1**
>
> We thank the reviewer for this important suggestion. To further substantiate the robustness of our results, we have supplemented additional experiments across multiple random seeds, and report the mean and standard deviation of the performance metrics. The results show consistently low variance, confirming the stability and reliability of SEER. We will include these statistics in the revised manuscript. Due to space limitations, please refer to the results at the **following link**: https://anonymous.4open.science/r/ICML-SEER-Rebutal-97F1/stability.md
>
> **Reply to W2 & Q1**
>
> We thank the reviewer for this question. Our experimental setup follows the evaluation framework of TFB. Specifically, the configurations and hyperparameters of baseline models are inherited from this widely recognized framework, which provides well-established and carefully tuned implementations for fair comparison. By using the officially released scripts and default settings within this framework, we ensure that all baselines are evaluated under consistent and well-optimized conditions, thereby maintaining the fairness and reliability of the comparison.
>
> **Reply to W3**
>
> We thank the reviewer for pointing this out. We will update all outdated references by replacing ArXiv versions with their corresponding official conference or journal publications in the revised manuscript.
>
> **Reply to W4 & Q2**
>
> We thank the reviewer for this insightful question. Although SEER is designed for robust time series forecasting, standard benchmark datasets such as ETT and Traffic are still highly relevant. As discussed in TFB, these commonly used datasets are not strictly stationary and often exhibit distribution shifts, meaning that certain subsequences may naturally contain low-quality segments. In addition, to more thoroughly evaluate robustness, we have conducted additional experiments on both synthetically corrupted datasets (with injected low-quality conditions) and real-world low-quality datasets. These complementary settings allow us to comprehensively validate the effectiveness of SEER under both standard benchmarks and more challenging scenarios.

---

> > ### Author Rebuttal · Reviewer_8qts · 2026-04-01
> >
> > The author answered my questions and My concerns have been adequately addressed.

---

> > > ### Author Response · Authors · 2026-04-04
> > >
> > > Dear Reviewer, we are thrilled that our responses have effectively addressed your questions and comments. We would like to express our sincerest gratitude for taking the time to review our paper and provide us with such detailed feedback!

---

### Official Review · Reviewer_ndN5 · 2026-03-09

**Soundness:** 3
**Presentation:** 4
**Significance:** 4
**Originality:** 4
**Overall Recommendation:** 5
**Confidence:** 4

**Summary:**

This paper proposes SEER, a transformer-based forecasting framework designed to enhance robustness against low-quality time series exhibiting missing values, anomalies, white noise, and distribution shifts. Extensive experiments on standard multivariate and large-scale univariate benchmarks, as well as on synthetic and real-world degraded datasets, demonstrate consistent improvements over strong baselines, indicating that SEER enhances both forecasting accuracy and robustness.

**Compliance With Llm Reviewing Policy:**

Affirmed.

**Final Justification:**

The paper targets robust long-horizon forecasting under low-quality inputs, with clear motivation and comprehensive experiments. The method is general, and presentation is clear.
The authors have satisfactorily addressed all concerns: clarifying how τ is chosen, detailing ablation variants, referencing dataset descriptions in the appendix, and justifying causal self-attention. All issues can be fixed in a minor revision.
I therefore choose to accept.

**Key Questions For Authors:**

Please see the weaknesses. If my concerns are adequately addressed, I will increase my score.

**Limitations:**

Future work could explore methods to automate the determination of the filtering threshold $\tau$, potentially adapting it dynamically to different datasets or scenarios, thereby enhancing the robustness and broader applicability of the proposed method. This is intended as a discussion point rather than a criticism.

**Strengths And Weaknesses:**

Strengths:

1. The evaluation is extensive, spanning 8 multivariate datasets and 8,068 univariate series, along with robustness assessments under four types of synthetic corruptions and selected real-world degraded datasets.

2. Robust long-horizon forecasting under low-quality inputs is a timely and meaningful problem; the proposed approach is general in design and appears readily compatible with mainstream transformer-based pipelines.

3. The figures and tables are clear, well-organized, and easy to interpret.

4. The core motivation is well-defined and compelling, addressing a significant limitation of existing methods.



Weaknesses:

1. How is the token selection threshold τ determined for each dataset and forecasting horizon? Is it fixed globally or tuned on validation sets? Additionally, how sensitive is the model’s performance to this hyperparameter in practice?

2. The textual description of the ablation study design is relatively brief and could be elaborated to clarify how each component is systematically removed or modified.
3. The rationale behind the selection of real-world low-quality datasets, as well as their descriptions, is relatively brief. It is recommended to further elaborate on this part so that readers can more clearly understand the experimental setup and details.
4. In (d) Replaced Causal Self-Attention, why is Causal Self-Attention adopted to model interactions among tokens? Would standard Self-Attention with positional encoding also be sufficient to prevent token representations from collapsing into identical embeddings? What is the specific motivation for choosing Causal Self-Attention in this design?

---

> ### Author Rebuttal · Authors · 2026-03-30
>
> Dear Reviewer ndN5, thanks for your detailed suggestions.
>
> **Reply to W1 & Limitations**
>
> We thank the reviewer for this important question. The token selection threshold $\tau$ is treated as a dataset-dependent hyperparameter, and is selected based on validation performance for each dataset. We would like to emphasize that $\tau$ reflects a fundamental trade-off: a non-zero threshold helps filter out noisy or unreliable tokens, while an overly large threshold may discard useful information, leading to potential information loss. As a result, the performance with respect to $\tau$ does not necessarily follow a monotonic trend. In practice, the optimal value of $\tau$ depends on the characteristics of each dataset, such as noise level and data quality. We will clarify this selection strategy in the revised manuscript. In addition, exploring adaptive or automated mechanisms for determining $\tau$ is an important direction for future work.
>
> **Reply to W2**
>
> We thank the reviewer for this question. Due to space limitations, the original description of the ablation study was concise. In the revised manuscript, we provide a clearer and more detailed explanation of each variant as follows:
>
> - **w/o MoE**: The Augmented Embedding module removes the Augmented Patch Embedding and only retains the standard patch embedding.
> - **Simple Series Embedding**: The Augmented Series Embedding is replaced with a simplified series embedding as used in iTransformer.
> - **w/o Token Filter**: The Automated Token Filter is disabled by setting $\tau = 0$, which retains all tokens instead of performing patch-wise filtering.
> - **w/o Feature Reduction**: The dimensionality reduction operation in Eq. (16) is removed.
>
> **Reply to W3**
>
> We thank the reviewer for this helpful suggestion. Due to space limitations, the detailed descriptions of the selected real-world low-quality datasets and the rationale behind their selection were not included in the main text. Instead, this information is provided in Appendix C.3 of the original manuscript.
>
> **Reply to W4**
>
> We thank the reviewer for this insightful question. The use of Replaced Causal Self-Attention (RCSA) is motivated by the fact that multiple time positions may be replaced by the same global prototype tokens in our framework. In this setting, the replaced causal masking mechanism ensures that each token only attends to its own preceding context, thereby preserving temporal order and enabling position-specific representations. As a result, even when the input tokens are identical, their resulting hidden representations remain highly distinguishable, while maintaining temporal continuity across patches. Standard Self-Attention with positional encoding can partially alleviate this issue. However, in our empirical evaluations, it consistently underperforms compared to RCSA.

---

> > ### Author Rebuttal · Reviewer_ndN5 · 2026-04-02
> >
> > Thank you for the detailed response and clarifications. I believe the authors have addressed my questions and concerns satisfactorily.

---

> > > ### Author Response · Authors · 2026-04-04
> > >
> > > Dear Reviewer, we are thrilled that our responses have effectively addressed your questions and comments. We would like to express our sincerest gratitude for taking the time to review our paper and provide us with such detailed feedback!

---

### Official Review · Reviewer_b4EN · 2026-03-13

**Soundness:** 3
**Presentation:** 3
**Significance:** 3
**Originality:** 3
**Overall Recommendation:** 4
**Confidence:** 5

**Summary:**

This paper proposes SEER, a Transformer-based model for robust time-series forecasting under imperfect observations. It combines an Augmented Embedding Module for improving patch representations with a Learnable Patch Replacement Module that filters and replaces unreliable patches using global series-wise tokens and causal attention. The paper evaluates SEER on multiple forecasting benchmarks and robustness settings to demonstrate its effectiveness.

**Compliance With Llm Reviewing Policy:**

Affirmed.

**Final Justification:**

**Thank you to the authors for the detailed rebuttal and the additional experiments.** My main concerns have been substantially addressed, and I therefore raise my score from 3 to 4.

However, the clarification provided in response to Q1 should be incorporated into the revised manuscript rather than remaining only in the rebuttal. To avoid possible misunderstanding, I encourage the authors to state this point more clearly in the paper, as it is important for understanding the mechanism of the method.

**Key Questions For Authors:**

1. How do the authors justify treating missing values, distribution shifts, anomalies, and white noise as a single category of “low-quality situations,” given that anomalies and distribution shifts may sometimes carry predictive signal or require adaptation rather than suppression?

2. Since the proposed mechanism does not explicitly identify whether a patch is problematic due to missingness, anomalies, white noise, or shift, how do the authors justify interpreting it as explicitly handling these different non-robust situations?

3. Could the authors provide stronger evidence that the observed gains come specifically from the proposed robustness mechanism, rather than from the combined effect of multiple components such as embedding design, feature reduction, and replacement?

4. Could the authors include direct ablations to validate the necessity of Replaced Causal Self-Attention and the additional standard MSA after replacement?

5. How do the authors explain the robustness results for distribution shift and threshold sensitivity, especially given that Appendix Table 6 does not show a clearly increasing relative advantage under stronger shifts and Figure 5d shows weak or unstable trends on ETTm2 and ETTh2?

6. Could the authors clarify how the real-world missing-value datasets were preprocessed or adapted to make them compatible with the forecasting models used in the main comparison, given that these datasets appear closer to irregular multivariate settings?

**Limitations:**

yes

**Strengths And Weaknesses:**

### Soundness
- The overall framework is technically coherent. The combination of augmented embeddings, token filtering/replacement, and attention-based refinement is conceptually reasonable.
- The empirical evaluation is fairly broad, including standard forecasting benchmarks, robustness experiments, ablations, and sensitivity studies.
- The central robustness framing is not fully convincing. The paper treats missing values, distribution shifts, anomalies, and white noise as a unified category of “low-quality situations,” but this assumption is not always fully justified. Among these factors, missing values are relatively clear nuisances, whereas anomalies and distribution shifts are not necessarily harmful. In many real applications, they may themselves carry predictive signal or indicate regime changes that the model should adapt to rather than suppress. For this reason, it is unclear whether filtering or replacing such patches should always be interpreted as “robust modeling.” A more adaptive perspective, similar in spirit to methods such as RevIN[1] for handling distribution shift, may be more appropriate than implicitly treating the shift itself as something to remove.
- More fundamentally, the proposed mechanism does **not explicitly identify** whether a patch is problematic because of missingness, anomalies, white noise, or shift. Instead, it learns a generic score and filters/replaces patches that appear less useful for prediction. As a result, the paper’s interpretation that the method explicitly tackles these different non-robust situations relies on an implicit assumption that is not directly validated.
- The current experiments do not sufficiently establish that the gains come specifically from the proposed robustness mechanism. The ablation study suggests that the improvements are not cleanly attributable to the token filter alone. In particular, on datasets such as ETTh2 and ETTm2, removing feature reduction causes larger degradation than removing the token filter. More broadly, the gains appear to come from the **combination** of embedding design, feature reduction, replacement, and other components, rather than clearly from the robustness-oriented filtering mechanism alone.
- The paper does not directly validate some important architectural choices. Since the paper explicitly motivates Replaced Causal Self-Attention through the claim that each patch should attend only to preceding patches and the prepended global prototype, this design choice should be supported by direct ablations. Likewise, the necessity of the additional standard MSA after replacement is not directly demonstrated.
- The robustness evidence for **distribution shift** is weaker than the narrative suggests. In Appendix Table 6, SEER does not show a clearly increasing relative advantage as the shift becomes more severe. If the method truly targeted distribution shift effectively, one might expect its advantage to become more pronounced under stronger shifts[1].
- The **missing-value benchmark** is also somewhat questionable. The real-world datasets used there appear closer to irregular multivariate settings, and it is not clearly explained what preprocessing or adaptation was applied to make them compatible with the forecasting models used in the main comparison.
- Figure 5d is also somewhat puzzling. Since increasing the threshold leads to more aggressive filtering/replacement, one might expect a clearer trend if such filtering were especially beneficial on noisy datasets. However, ETTm2 is largely insensitive to the threshold, while ETTh2 shows a non-monotonic and unstable pattern, with performance tending to degrade as filtering becomes more aggressive.

### Presentation
- The paper is generally well written and easy to follow.
- The architecture is decomposed into understandable parts, which helps readability.
- In the abstract, the statement that existing patch-based methods fail to dynamically select patches is overstated, since prior work such as SRSNet has already explored adaptive patch selection/reassembly. A more accurate framing would be to clarify **which specific limitation of prior patch-based methods SEER addresses**, rather than implying that dynamic patch selection itself is new.
- Some parts of the Main Results section make claims that go beyond what the experiments directly establish. In particular, several paragraphs interpret performance gains as if they could be causally attributed to specific modules, even though the current evidence is not strong enough to support such direct attribution.
- There are also minor presentation issues, such as template/title formatting, that should be cleaned up in the final version.

### Significance
- Robust forecasting under imperfect real-world data is an important and practically meaningful problem across many domains.
- The Augmented Embedding Module is a potentially useful contribution at the representation level and offers a somewhat different perspective on patch embeddings.
- However, the current evidence does not fully establish that the proposed method specifically improves forecasting under genuinely non-robust conditions, as opposed to simply providing a stronger overall architecture.

### Originality
- The Augmented Embedding Module appears to be a meaningful contribution at the input-representation level.
- Replacing low-utility local patches with a global prototype and then recontextualizing them is an interesting architectural idea.
- However, while the paper includes interesting design choices, its central originality claim on robust forecasting under imperfect observations is not sufficiently validated.


[1] Kim, Taesung, et al. "Reversible instance normalization for accurate time-series forecasting against distribution shift." International conference on learning representations. 2021.

---

> ### Author Rebuttal · Authors · 2026-03-30
>
> **Reply to Concerns on Soundness**
>
> - （Also reply to Q1）We clarify that in SEER, “low-quality” data refers to non-structural artifacts from external factors (e.g., sensor failures or transmission errors), rather than intrinsic signal dynamics. We agree that in some domains (e.g., seismic data), anomalies or shifts are meaningful signals, and will explicitly clarify this distinction in the revised manuscript.
>
> - （Also reply to Q2）To validate SEER under diverse non-robust conditions, we conduct experiments on real-world datasets (Section 5.3.2, Table 5). SEER consistently outperforms baselines across all scenarios. We also provide visualization results to further demonstrate its ability to identify and mitigate different types of data corruption. Please refer to: https://anonymous.4open.science/r/ICML-SEER-Rebutal-97F1/visualization.md
>
> - We agree that the performance gains should not be attributed solely to the token filtering mechanism. As a unified framework, SEER benefits from the joint effect of multiple components, as evidenced by the ablation results, including embedding design, feature reduction, replacement, and filtering. We further note that ablations conducted on standard multivariate datasets may not fully highlight the effectiveness of the token filter. To better validate its role, we have additionally included ablation experiments on four types of real-world low-quality data. Please refer to: https://anonymous.4open.science/r/ICML-SEER-Rebutal-97F1/new_datasets_ablation.md
>
> - （Also reply to Q4）We add ablation studies to validate these design choices. Specifically, we remove the additional standard MSA to test its necessity, and replace the Replaced Causal Self-Attention with a standard MSA with positional encoding. The results provide direct evidence of the effectiveness of the proposed components. Please refer to: https://anonymous.4open.science/r/ICML-SEER-Rebutal-97F1/new_datasets_ablation.md
>
> - Upon re-examination, we found that the shift injection in Appendix Table 6 (Algorithm 4 in B4) was overly extreme, with an excessively large segment length $L_{\text{seg}}$, leading to unrealistic deviations from the original semantics. We have adjusted $L_{\text{seg}}$ to a more reasonable range and re-conducted the experiments. The updated results show that SEER achieves more pronounced gains as the degree of distribution shift increases.  Please refer to: https://anonymous.4open.science/r/ICML-SEER-Rebutal-97F1/new_shifting.md
>
> - （Also reply to Q6）The real missing-value data indeed falls under irregular multivariate time series, as irregularity inherently involves the presence of missing observations. Our processing pipeline follows the HyperIMTS framework.
>
> - （Also reply to Q5）We would like to clarify that the threshold reflects a classic trade-off: a non-zero threshold helps filter out noisy or unreliable tokens, while an excessively high threshold may discard useful signal and lead to information loss. Therefore, the relationship is not necessarily expected to be monotonic. In practice, the optimal threshold depends on the specific characteristics of each dataset.
>
> **Reply to Concerns on Presentation**
>
> - In SRSNet, the main branch still processes all patches uniformly, while the selection branch primarily focuses on patch reassembly and does not explicitly handle negatively impacted patches. In contrast, the key distinction of SEER lies in its robustness-oriented design: it explicitly identifies and filters negatively impacted patches via an automatic token filtering mechanism, and further leverages a global sequence-level token for semantic replacement to mitigate the effect of low-quality data. We will revise the abstract accordingly to provide a more precise positioning, clarifying that SEER addresses the specific limitation of negative patch suppression and global semantic recovery, rather than implying that dynamic patch selection itself is a novel contribution.
>
> - In the revised manuscript, we will refine our language to be more rigorous, shifting from direct causal claims to a more nuanced description of the correlations observed between specific modules and performance gains.
>
> - We will carefully review and revise the manuscript to fully comply with the ICML 2026 formatting guidelines, ensuring a professional presentation in the final version.
>
> **Reply to Concerns on Significance & Originality**
>
> - （Also reply to Q3）To validate that SEER effectively handles these diverse non-robust conditions, we conducted comparative experiments on representative real-world datasets as detailed in Section 5.3.2 and Table 5. The results demonstrate that SEER consistently achieves superior performance over baseline methods across all four scenarios.  To better validate the key modules role, we have additionally included ablation experiments on four types of real-world low-quality data.  Please refer to: https://anonymous.4open.science/r/ICML-SEER-Rebutal-97F1/new_datasets_ablation.md

---

> > ### Author Rebuttal · Reviewer_b4EN · 2026-04-02
> >
> > **Thank you to the authors for the detailed rebuttal and the additional experiments.**
> >
> > Overall, I believe that Q3 and Q4 have been addressed, and my concerns regarding significance, originality, and presentation have also been largely resolved.
> >
> > However, a few questions still remain.
> >
> > - Q1. Thank you for clarifying that “low-quality” refers to external, non-structural artifacts rather than intrinsic signal dynamics. However, this raises a follow-up question: does SEER have any mechanism to distinguish whether anomalies or shifts are meaningful signals or merely external corruption? Or is this distinction left to implicit learning by the model?
> >
> > - Q2. My original concern was slightly different from the current experimental setup. The current results mainly show robustness to injected noise or perturbations. What I would like to see is whether SEER also works well under real distribution shift naturally occurring in the data, without artificial modification. For example, it would be helpful to evaluate cases where the sequence follows one pattern and then transitions to a different regime, along with qualitative visualization.
> >
> > - Q5. I am still not fully convinced by the explanation of the distribution-shift experiment. If the original setting was “too extreme,” it remains unclear why it was inappropriate and how the revised setting was changed. More importantly, if SEER is truly shift-robust, I would expect a clearer explanation of why its relative advantage does not become more pronounced as shift severity increases. The claim that the original perturbation was so extreme that it “broke the semantics” also seems to suggest a limitation of the claimed robustness.
> > - Q6. The clarification on the irregular missing-value benchmark is still insufficient. Irregular multivariate time series differ fundamentally from regular time series because observations occur at non-uniform timestamps, often with different sampling times across variables. As a result, performance depends heavily on how such irregularity is handled in preprocessing or model design; otherwise, methods that assume uniformly sampled inputs may not be fairly evaluated. The rebuttal states that the pipeline follows HyperIMTS, but HyperIMTS is based on a GNN-style irregular-time setting. Therefore, the paper should clearly explain how SEER, as well as the regular-time MLP/Transformer baselines, are adapted to this setting: specifically, how irregular observations are converted into model-compatible inputs, how timestamps or time gaps are represented, and whether the same adaptation is applied consistently across baselines. Without these details, it is difficult to assess the validity and fairness of the comparison.
> >
> > If these remaining points are clarified, I would be willing to increase my score.

---

> > > ### Author Response · Authors · 2026-04-02
> > >
> > > We apologize for the limited space in our previous response, and we thank the reviewer for the opportunity to further clarify these important points.
> > >
> > > **Reply to Q1**
> > >
> > > The distinction is indeed left to implicit learning by the model. SEER is trained end-to-end with the forecasting objective, and thus learns to identify *patches that are harmful to prediction*.
> > >
> > > Concretely, if an abrupt change is informative for forecasting, retaining it will reduce the prediction loss, and the learned filter will preserve it. In contrast, only those patches that consistently increase the loss (e.g., corrupted or misleading signals) are suppressed.
> > >
> > > **Reply to Q2**
> > >
> > > In addition to robustness under injected noise or perturbations, we also evaluate SEER on *real-world distribution shift datasets*.
> > >
> > > Specifically, as shown in Table 5 (Section 5.3.2), the ILI and NASDAQ datasets are identified as the most distribution-shifted datasets according to the shifting metric defined in TFB. The shifting patterns shown in the visualization results, which we provide at https://anonymous.4open.science/r/ICML-SEER-Rebutal-97F1/visualization.md, are derived directly from the real-world NASDAQ dataset.
> > >
> > > Both the quantitative results and visualizations demonstrate SEER’s effectiveness in handling naturally occurring distribution shifts. We will clarify this more explicitly in the revision.
> > >
> > > **Reply to Q5**
> > >
> > > We sincerely apologize for the insufficient explanation due to space limitations in the previous response, and we provide a more detailed clarification here.
> > >
> > > In the original setting (Appendix Algorithm 4), the shifting perturbation was applied over long segments across the entire dataset. Specifically, due to the large temporal span $T$, the segment length $L_{seg}$ was also large, and shifting was applied to all segments of length $L_{seg}$ throughout the training data. This led to long-term distortion of local statistics (e.g., local mean), which severely obscured meaningful temporal patterns and degraded the quality of the training data. Such perturbations are not consistent with realistic distribution shifts.
> > >
> > > To address this issue, we revised the design to apply *localized and controlled shifting*. Concretely, in the updated algorithm (line 9):
> > >
> > > $$
> > > X_{\text{noisy}}[\text{end}-t : \text{end}, c] \leftarrow X_{\text{noisy}}[\text{end}-t : \text{end}, c] + \delta
> > > $$
> > >
> > > where $t$ is a parameter that controls the length of the shifted segment. This ensures that the perturbation remains local while preserving the overall temporal structure, making it more aligned with real-world distribution shifts. The updated results show that SEER achieves more pronounced gains as the degree of distribution shift increases. Please refer to: https://anonymous.4open.science/r/ICML-SEER-Rebutal-97F1/new_shifting.md.
> > >
> > > **Reply to Q6**
> > >
> > > To ensure strict fairness, we rely on the standardized preprocessing pipeline natively provided by the PyOmniTS framework. While HyperIMTS is a GNN-based model, its official PyOmniTS framework was explicitly built as a *unified, extensible pipeline* to benchmark both irregular-specific models and standard regular-time models (e.g., MLPs, Transformers) on a level playing field.
> > >
> > > Specifically, for SEER and all regular-time baselines, the framework handles the irregular-to-regular adaptation uniformly through the following steps:
> > >
> > > 1. Conversion into Model-Compatible Inputs (Canonical Pre-Alignment): Irregular and unaligned observations across different variables are mapped onto a unified global temporal grid (typically the union of all unique timestamps within a given window). To construct the standard dense tensor format $\in \mathbb{R}^{T \times D}$ required by regular MLPs/Transformers, missing values at these aligned timestamps are padded with zeros.
> > > 2. Representation of Timestamps and Time Gaps: To prevent the model from treating padded zeros as actual sensor readings, the framework simultaneously generates a corresponding binary observation mask (1 for observed, 0 for padded). This mask, alongside explicit temporal features (such as time-gap intervals or continuous timestamp embeddings), is appended/concatenated to the input. This ensures that regular models are explicitly aware of the irregular sampling intervals and the exact locations of the missing values.
> > > 3. Absolute Consistency Across Baselines: We emphasize that this exact same adaptation process—including the grid alignment, zero-padding, masking, and timestamp representation—is applied strictly and consistently to SEER and all regular-time baselines. They all receive identical pre-aligned tensors from the PyOmniTS data loaders.
> > >
> > > By utilizing this unified preprocessing wrapper, we ensure that the performance differences observed in our benchmark are purely attributable to the architectural designs of the models themselves, rather than any discrepancies in data handling. We will include this clarification in the revision to improve transparency and reproducibility.

---

### Decision · Program_Chairs · 2026-04-30

**Decision:**

Accept (regular)

**Comment:**

This paper proposes transformer-based robust time Series forecasting method for low-quality data issues including missing values, distribution shifts, anomalies, and white noise. The overall architecture comprises the augmented embedding, the automated token filtering, and the replaced causal attention. All of the reviewers are satisfied with the rebuttals from the authors, and most of them have raised their scores to support acceptance.

The authors are encouraged to provide stronger evidence including stronger ablation studies and case studies in the final version that each of the proposed sub-techniques could contribute to the robust time series modeling for different low-quality data issues.

Overall, this is a good paper, thus I recommend acceptance.